# The Latent Color Subspace: Emergent Order in High-Dimensional Chaos

**Mateusz Pach** [* 1 2 3]   **Jessica Bader** [* 1 2 3]   **Quentin Bouniot** [1 2 3 4]   **Serge Belongie** [5]   **Zeynep Akata** [1 2 3]

## Abstract

Text-to-image generation models have advanced rapidly, yet achieving fine-grained control over generated images remains difficult, largely due to limited understanding of how semantic information is encoded. We develop an interpretation of the color representation in the Variational Autoencoder latent space of FLUX.1 [Dev], revealing a structure reflecting Hue, Saturation, and Lightness. We verify our Latent Color Subspace (LCS) interpretation by demonstrating that it can both predict and explicitly control color, introducing a fully training-free method in FLUX based solely on closed-form latent-space manipulation. Code is available at https://github.com/ExplainableML/LCS.

## 1. Introduction

Flow Matching (FM) models are increasingly capable of generating high-quality, accurate images, enabling their use across a wide range of practical applications (Dinkevich et al., 2025; Yellapragada et al., 2024; Wang et al., 2025). Nonetheless, precise and reliable control over generated images remains a significant challenge, despite being essential for many of these applications. Prior work improved controllability for image generation (Zhang et al., 2023; Ye et al., 2023) and editing (Black Forest Labs, 2025b). However, these approaches often depend on additional models or training, increasing system complexity without substantially improving understanding of the underlying mechanisms. This lack of insight makes it difficult to establish trust in the system. Rather than increasing system complexity, we aim to develop a clearer interpretation of how FLUX.1 [Dev] (FLUX) (Black Forest Labs, 2024) processes a fundamental image component: color. To validate our interpretation, we

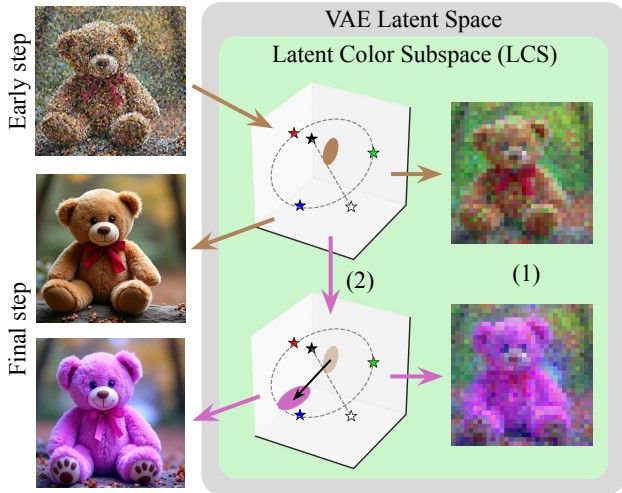

*Figure 1.* We find a simple color subspace in the VAE embedding space of FLUX which can be interpreted as cylindrical coordinates corresponding to Hue, Saturation, and Lightness, enabling (1) inexpensive observation and (2) targeted intervention.

want to show two key properties: it is (1) *accurate*, faithfully reflecting the final image's emerging features, and (2) *causal*, enabling deliberate intervention. Unfortunately, achieving such understanding in text-to-image (T2I) generation models is difficult due to deep learning's black-box nature, a challenge further compounded by the step-wise prediction process of T2I generation and its operation within the high-dimensional latent space of a variational autoencoder (VAE), which is itself largely uninterpretable.

Still, we develop and verify a simple interpretation of color in the VAE latent space of FLUX (Black Forest Labs, 2024). We observe that color occupies a three-dimensional subspace, forming a bicone-like structure that closely mirrors the Hue–Saturation–Lightness (HSL) representation. By combining this insight with an understanding of how image patches evolve across FM timesteps, we construct a functional interpretation of color in the latent space that generalizes across HSL colors. This allows color to be interpreted at intermediate timesteps directly in the latent space through lightweight transformations, without needing the 50-million-parameter VAE decoder. We validate the accuracy of our interpretation by using it to *observe* mid-generation color representations in the latent space and *intervene*, guiding the generation toward target colors. When combined with se-

*Equal contribution   [1]Technical University of Munich [2]Helmholtz Munich [3]Munich Center for Machine Learning [4]LTCI, Télécom Paris, Institut Polytechnique de Paris [5]University of Copenhagen. Correspondence to: Mateusz Pach <mateusz.pach@tum.de>.

*Proceedings of the 43rd International Conference on Machine Learning*, Seoul, South Korea. PMLR 306, 2026. Copyright 2026 by the author(s).

mantic segmentation, this intervention enables fine-grained control over the colors of specific objects (see Figure 1).

The primary contributions of this work are threefold: (1) To our knowledge, we are the first to show that color lives in a three-dimensional subspace of FLUX's VAE latent space, closely resembling HSL representation; (2) We leverage this understanding to develop a working interpretation of color encoding that generalizes across the full HSL color space; (3) We introduce a novel, entirely training-free localized color-intervention method that relies solely on a mechanistic understanding of FLUX's internal representations.

## 2. Related Works

The adoption of diffusion models (Rombach et al., 2022) has transformed T2I generation, typically operating in the latent space of a VAE (Kingma & Welling, 2014). These models have improved in image quality and prompt adherence (OpenAI, 2023; Midjourney, 2025; Google DeepMind, 2025; Podell et al., 2023; Chen et al., 2024), recently shifting toward transformer-based diffusion architectures (Peebles & Xie, 2023; Esser et al., 2024; Black Forest Labs, 2024; Wu et al., 2025) and taking on a FM perspective (Lipman et al., 2022; Albergo & Vanden-Eijnden, 2023; Liu et al., 2023). Despite these advances, fine-grained control remains limited across several dimensions, including pose and layout (Zhang et al., 2023), spatial positioning (Bader et al., 2025b), and color (Laria et al., 2026). These limitations have motivated work on controllable generation through optimization (Zhang et al., 2023; Eyring et al., 2024; 2025; Li et al., 2023; Farshad et al., 2023; Shum et al., 2025), though training-free approaches have also been explored (Bader et al., 2025a;b; Oorloff et al., 2025).

Despite rapid advances in T2I models, their underlying mechanisms remain less explored. Prior work has begun to uncover key internal processes, including why they generalize (Niedoba et al., 2025), how they generate spatial relations (Wang et al., 2026), and how biases emerge (Shi et al., 2025). Complementary approaches leverage attention mechanisms within T2I models to analyze or control generation (Chefer et al., 2023; Hertz et al., 2023; Tang et al., 2023), as well as sparse autoencoders to identify interpretable and intervenable directions in model representations (Kim et al., 2025b; Daujotas, 2024; Shabalin et al., 2025; Shi et al., 2025). Furthermore, attention mechanisms in DiT models have proven effective for segmentation (Kim et al., 2025a; Helbling et al., 2025; Hu et al., 2025).

Color control in FM models has been studied via color conditioning (Shum et al., 2025) and color–style disentanglement (Zhang et al., 2025). It can be enabled by learned color prompts (Butt et al., 2024), IP-Adapters (Laria et al., 2026), and inpainting or ControlNet-based approaches (Liu et al., 2025). These methods increase model complexity without improving interpretability, whereas ours leverages understanding to enable control. Others focus on color control in editing (Liang et al., 2025; Vavilala et al., 2025; Yang et al., 2025). Concurrent work analyzes color encoding in the VAE latent space (Arias et al., 2025) but is more limited, lacking prediction, intervention, and temporal FM analysis.

## 3. Analysis of Color in FM VAE Space

To develop an interpretation of color in FLUX, we must understand how color is represented in the VAE space and how this space is traversed during the denoising process.

### 3.1. Preliminaries

**Variational Autoencoder**    Modern FM models are typically trained in a compressed VAE latent space. For an input image $\mathbf{x}$, the encoder $E_\phi$ defines a posterior

$$q_\phi(\mathbf{z} \mid \mathbf{x}) = \mathcal{N}\big(\mathbf{z}; \boldsymbol{\mu}(\mathbf{x}), \mathrm{diag}(\boldsymbol{\sigma}(\mathbf{x})^2)\big).$$

Latent variables are sampled via the reparameterization trick

$$\mathbf{z} = \boldsymbol{\mu}(\mathbf{x}) + \boldsymbol{\sigma}(\mathbf{x}) \odot \boldsymbol{\epsilon}, \quad \boldsymbol{\epsilon} \sim \mathcal{N}(\mathbf{0}, \mathbf{I}).$$

The decoder reconstructs the image as $\hat{\mathbf{x}} = D_\theta(\mathbf{z})$.

Training combines reconstruction, regularization, and adversarial objectives. The perceptual reconstruction loss $L_{\mathrm{rec}}$ is defined with LPIPS (Zhang et al., 2018), while $L_{\mathrm{reg}}$ enforces alignment of $q_\phi(\mathbf{z} \mid \mathbf{x})$ with a unit Gaussian prior via KL divergence. To improve realism, a patch-based discriminator $D_\psi$ is trained to distinguish real and reconstructed images, yielding an adversarial loss $L_{\mathrm{adv}}$. The full objective is:

$$\min_{\phi,\theta} \max_{\psi} \ \mathbb{E}_{\mathbf{x}}\Big[ L_{\mathrm{rec}} + \beta L_{\mathrm{reg}} + \lambda_{\mathrm{adv}} L_{\mathrm{adv}} \Big],$$

where $\beta$ and $\lambda_{\mathrm{adv}}$ are weighting factors.

**Flow Matching**    FLUX is trained with FM objective. Let $\mathbf{z}_0 \sim \mathcal{N}(\mathbf{0}, \mathbf{I})$ denote an initial noise sample in latent space and $\mathbf{z}_1$ a clean latent encoding of an image obtained from the VAE. FM learns a continuous-time velocity field $\mathbf{v}_\theta(\mathbf{z}, t)$ that transports samples from the noise distribution to the data distribution by minimizing

$$\mathcal{L}_{\mathrm{FM}} = \mathbb{E}_{t \sim \mathcal{U}(0,1),\, \mathbf{z}_t}\left[ \left\| \mathbf{v}_\theta(\mathbf{z}_t, t) - \frac{d\mathbf{z}_t}{dt} \right\|^2 \right],$$

where the interpolation path is defined as

$$\mathbf{z}_t = (1-t)\, \mathbf{z}_0 + t\, \mathbf{z}_1, \quad \frac{d\mathbf{z}_t}{dt} = \mathbf{z}_1 - \mathbf{z}_0.$$

In practice, the model predicts the velocity $\mathbf{z}_1 - \mathbf{z}_0$ from an interpolated latent $\mathbf{z}_t$ and timestep $t$, conditioned on text embeddings. At inference, the learned velocity field is integrated with a numerical solver (Euler discretization in our case) to transport pure noise to a clean latent sample.

## 3.2. Color Representation in the VAE Space

To explore VAE-space color representation, we use $N = 400$ solid-color images, sampled uniformly from Hue–Saturation–Value (HSV) space. Each image $n$ is encoded with FLUX's VAE encoder, producing latents $\mathbf{Z}^n \in \mathbb{R}^{L \times d}$, where $L$ is the number of patches and $d$ is patch dimensionality. We average each image's $L$ patches, obtaining a single latent vector $\bar{\mathbf{z}}^n \in \mathbb{R}^d$. Applying PCA to these $N$ latent vectors, after centering by their mean $\boldsymbol{\mu} \in \mathbb{R}^d$, reveals that the first three principal components (PC) $\mathbf{B} \in \mathbb{R}^{d \times 3}$ account for $100\%$ of the variance, indicating that color information is confined to a 3D subspace of the VAE latent space. We refer to this subspace as the **Latent Color Subspace (LCS)**.

To understand LCS structure, we project the averaged latents $\bar{\mathbf{z}}^n$ into this subspace, yielding the average *color coordinates*

$$\bar{\mathbf{c}}^n = \mathbf{B}^\top (\bar{\mathbf{z}}^n - \boldsymbol{\mu}) \in \mathbb{R}^3, \qquad n \in 1, \ldots, N.$$

These coordinates reveal well-organized geometry (Figure 2). The first dimension spans light to dark, while the second and third jointly form a circular hue structure, with radius encoding saturation. Together, this geometry closely resembles the HSL color representation, organized in a bicone. This represents hue as an angle, saturation as the distance from the center, and lightness as an axis.

Beyond FLUX.1, we repeat these experiments on SD3.5 (Esser et al., 2024), FLUX.2 (Black Forest Labs, 2025a), and Qwen-Image's (Wu et al., 2025) VAEs. Across all models (see Appendix Section L), we observe similar color organization, providing preliminary evidence that the LCS is not specific to FLUX.1 but may occur more broadly across architectures.

## 3.3. Development of Color over Time During Diffusion

Next we analyze how color representations change over time in the FM model with the prompt "yellow and blue checkered tiles". We project the latent representations from various steps into the LCS, focusing on the hue dimensions (Figure 3a). Each latent patch is a dot in its eventual color, with six colored stars as reference points from $t = 50$. Latent patches start as a centered, color-mixed Gaussian and gradually cluster toward blue, yellow, and brown, showing smooth evolution toward the final colors from early steps.

To quantify the expected position of latent patches at each timestep given the final color, we generate 26 plain images of differently colored walls with the prompt "{$color$} wall" (see Appendix A for full color list). We then project each timestep's latents into the LCS, represent the image with the average of its patches, and visualize the results in Figure 3b. We observe vectors gradually moving outward from the origin, as the FM distribution requires latent patches to start near mid-grey and traverse dimensions that happen to

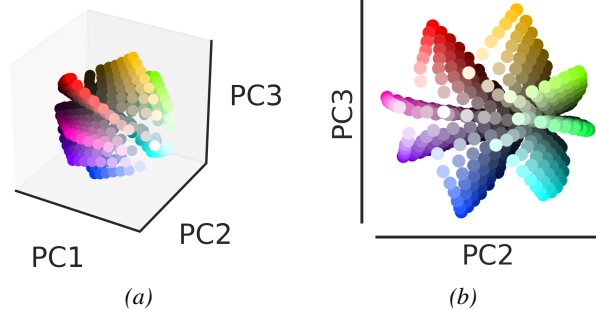

*(a)*              *(b)*

*Figure 2.* PCA shows color organization in the VAE latent space mirrors HSL: Hue forms a circle on the PC2–PC3 plane, Saturation is distance from the black-white axis, and Lightness lies on PC1.

mainly correspond to saturation and lightness in the VAE latent space as they move to their final color. Hence, in the context of the FM model, we interpret these dimensions as additionally relating to the timestep: more specifically, the timestep determines how far along its trajectory toward the final point a latent patch has progressed.

To capture a color's expected LCS position at timestep $t$, we must account for the distribution's time-dependent dynamics, independent of the generated colors. To this end, for each $t$ we compute two statistics: shift $\boldsymbol{\alpha}_t \in \mathbb{R}^3$ and per-axis scale $\boldsymbol{\beta}_t \in \mathbb{R}^3$ describing movement and expansion. We use the 26 images $\{X_i\}_{i=1}^{26}$ from the earlier qualitative analysis and project their token-averaged latents $\bar{\mathbf{z}}_t^i$ from timestep $t$ into LCS. The shift is computed as mean over images $\boldsymbol{\alpha}_t = \frac{1}{N} \sum_{i=1}^N \bar{\mathbf{z}}_t^i$ and scales as mean magnitudes after centering at $\boldsymbol{\alpha}_t$, that is $\boldsymbol{\beta}_t = \frac{1}{N} \sum_{i=1}^N |\bar{\mathbf{z}}_t^i - \boldsymbol{\alpha}_t|$. We report the values of these statistics in the Appendix.

## 4. Using the Latent Color Subspace

From our analysis in Section 3.2, we assume there exists a bijection between LCS and HSL. We propose an approximation of this mapping from a small set of known correspondences. We evaluate this approximation in two ways: observation and intervention. We show how to observe color directly in latent space mid-generation and intervene on latent representations to achieve a target HSL color.

### 4.1. Mapping Between Latent Color Subspace and HSL

We construct an approximation of assumed bijective mapping between the LCS coordinates $\mathbf{c} \in \mathbb{R}^3$ and HSL coordinates $(h, s, l)$ using a small set of canonical anchors $\mathcal{A} = \{\mathbf{h}_0, \ldots, \mathbf{h}_5, \mathbf{b}, \mathbf{w}\}$. The anchors correspond to six hues (red, blue, green, magenta, cyan, yellow) and black/white extremes. They are obtained by encoding plain color images into VAE latent space and projecting into LCS.

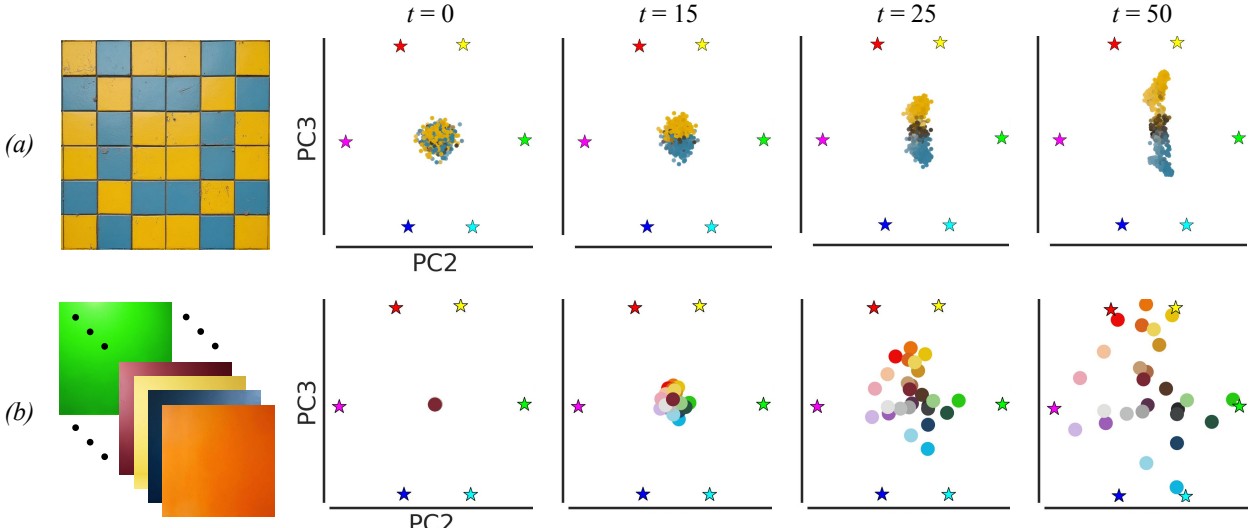

*Figure 3.* Flow Matching introduces an additional layer of complexity to our interpretation, as latents traverse the space over timesteps to reach their final destination. (a) In the Latent Color Subspace (LCS), colors evolve over timesteps $t$, starting mixed at the center and gradually moving toward their final positions. Dots represent individual patches, indicated in their ultimate colors, while stars orient the space with known color locations at $t = 50$. (b) Despite variation in individual patches, the *expected* relative position between colors stays consistent over timesteps in the LCS, but scaled with time. Shown on per-image averaged patches (circle) of 26 single-colored images.

**Decoding LCS → HSL**  Let $\mathbf{a} := \mathbf{w} - \mathbf{b}$ be the achromatic axis determined by the black and white anchors, and $\mathbf{c}_L$ be the projection of a point $\mathbf{c}$ onto this axis. Then, the lightness $l$ is the distance from $\mathbf{b}$ to $\mathbf{c}_L$ normalized by the length of $\mathbf{a}$.

$$l = \frac{\|\text{proj}_{\mathbf{a}}(\mathbf{c} - \mathbf{b})\|}{\|\mathbf{a}\|}, \qquad \mathbf{c}_L = \mathbf{b} + \text{proj}_{\mathbf{a}}(\mathbf{c} - \mathbf{b}),$$

The anchors $\mathbf{h}_0, \ldots, \mathbf{h}_5$ roughly form a circle and associate angular coordinates around the axis $\mathbf{a}$ with known HSL hue values. Hue $h$ can be angularly interpolated on this circle after removing the lightness component from the point $\mathbf{c}$.

Let $\mathbf{o} = \frac{1}{6} \sum_{i=0}^{5} \mathbf{h}_i$ be the approximate center of the hue anchor circle. Projecting $\mathbf{c}$ onto this circle gives

$$\mathbf{c}_H = \mathbf{c} + (\mathbf{o} - \mathbf{c}_L).$$

Select $\mathbf{h}_k$ and $\mathbf{h}_{k+1}$ as the neighboring anchors whose sector contains $\mathbf{c}_H - \mathbf{o}$, then the interpolation parameter is

$$\alpha = \frac{\angle(\mathbf{c}_H - \mathbf{o}, \ \mathbf{h}_k - \mathbf{o})}{\angle(\mathbf{h}_{k+1} - \mathbf{o}, \ \mathbf{h}_k - \mathbf{o})},$$

where $\angle(\mathbf{x}, \mathbf{y})$ is the angle between vectors $\mathbf{x}$ and $\mathbf{y}$. If $\theta_i$ is the HSL hue angle associated with anchor $\mathbf{h}_i$, then

$$h = \theta_k + \alpha(\theta_{k+1} - \theta_k).$$

Saturation $s$ is the distance from the achromatic axis normalized by the maximum attainable distance (MAD) at the same lightness. We assume the hue anchor circle forms the

equator of a bicone centered on the achromatic axis. Under this model, the MAD decreases linearly toward the poles at $l = 0$ and $l = 1$, and so saturation becomes

$$s = \frac{\|\mathbf{c} - \mathbf{c}_L\|}{R\left(1 - |2l - 1|\right)}, R = (1-\alpha)\|\mathbf{h}_k - \mathbf{o}\| + \alpha\|\mathbf{h}_{k+1} - \mathbf{o}\|,$$

and $R$ is the MAD for the hue of $\mathbf{c}$ at the equator's lightness. Together, this defines the decoding function $D$:

$$(h, s, l) = D(\mathbf{c}).$$

**Encoding HSL → LCS**  Given $(h, s, l)$, we reconstruct $\mathbf{c}$ with the same geometric model as decoding. The lightness determines point's $\mathbf{c}_L$ position on the achromatic axis:

$$\mathbf{c}_L = \mathbf{b} + l\,\mathbf{a}.$$

To recover $\mathbf{c}_H$, we interpolate spherically to find hue direction and linearly to adjust the distance from the achromatic axis according to the saturation. To interpolate the angle, we find $\mathbf{h}_k, \mathbf{h}_{k+1}$ such that $\theta_k \leq h \leq \theta_{k+1}$ and compute

$$\alpha = \frac{h - \theta_k}{\theta_{k+1} - \theta_k}.$$

With normalized anchor directions $\mathbf{d}_i = \frac{\mathbf{h}_i - \mathbf{o}}{\|\mathbf{h}_i - \mathbf{o}\|}$ and $\psi = \angle(\mathbf{d}_k, \mathbf{d}_{k+1})$ we spherically interpolate hue direction:

$$\mathbf{d}_H = \frac{\sin\left((1 - \alpha)\psi\right)}{\sin\psi}\mathbf{d}_k + \frac{\sin(\alpha\psi)}{\sin\psi}\mathbf{d}_{k+1}.$$

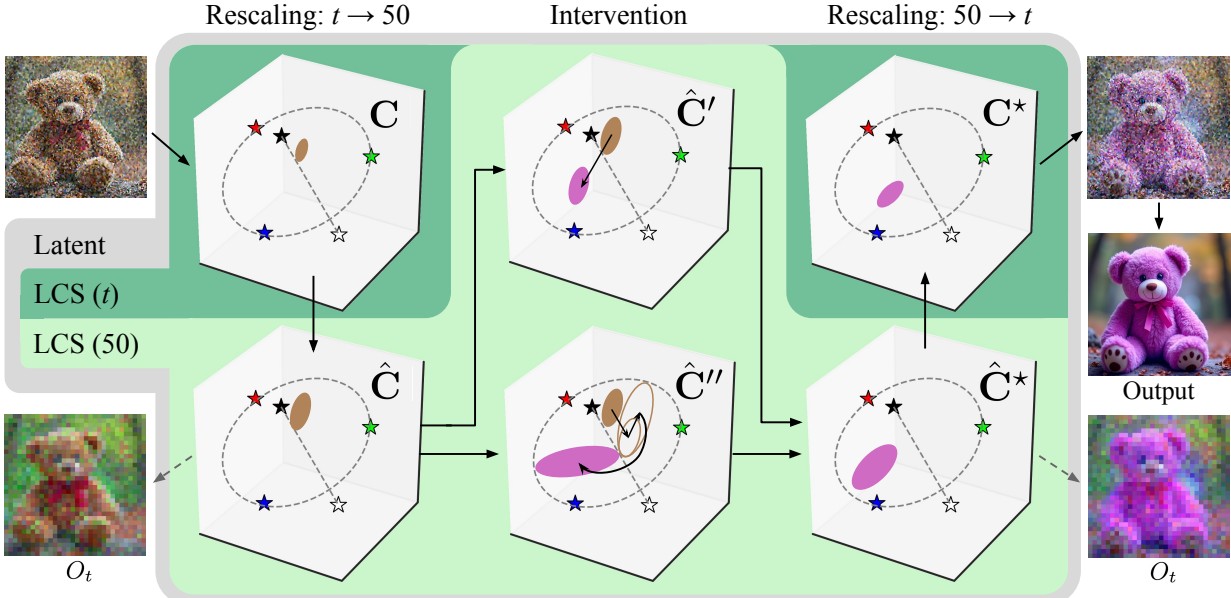

*Figure 4.* The Latent Color Subspace (LCS) enables observation and intervention during generation. At intermediate timestep $t$, we project the mid-generated sample from the FM VAE latent space (⬜) into the LCS (🟩) obtaining coordinates $\mathbf{C}$ and rescaling them to $\hat{\mathbf{C}}$, which matches timestep $t = 50$ statistics (🟩). Type II intervention ($\hat{\mathbf{C}}''$) modifies color by shifting, scaling, rotating to match the lightness, saturation, and hue respectively, while Type I intervention ($\hat{\mathbf{C}}'$) directly shifts to adjust all three. The interventions are interpolated to get $\hat{\mathbf{C}}^{\star}$ and rescaled back to timestep $t$ ($\mathbf{C}^{\star}$). Finally $\mathbf{C}$ is replaced with $\mathbf{C}^{\star}$ in the latent of the generated sample. With a simple projection into the LCS and the correct scaling, we can directly observe color ($O_t$) without the computationally heavy VAE decoder.

Using the same relations as before, $\mathbf{c}_H$ is calculated by scaling $\mathbf{d}_H$ by saturation, MAD $R$, and distance from the equator, and binding the vector to the center of the equator

$$\mathbf{c}_H = sR(1 - |2l - 1|)\,\mathbf{d}_H + \mathbf{o}.$$

Adding the lightness vector gives the encoding function:

$$\mathbf{c} = \mathbf{c}_H + (\mathbf{c}_L - \mathbf{o}) = E(h, s, l).$$

Taken together, $D$ and $E$ can approximate a mapping $\mathbf{c} \leftrightarrow (h, s, l)$, providing access to an interpretable, and well-organized LCS hidden inside the model.

### 4.2. Mid-Generation Color Observation

We can observe the colors which model is most likely to generate in the final image directly from LCS-projected latent $\mathbf{C} = [\mathbf{c}_i]_{i=1}^{L} \in \mathbb{R}^{L \times 3}$ at timestep $t$ with our decoding function $D$. However we have to remember that $D$ is defined in the default VAE latent space (i.e., at the final timestep), and in Section 3.3 we have shown that the statistics of the distribution in LCS change over time. Hence, we start by computing the coordinates normalized to timestep $t_{50}$:

$$\hat{\mathbf{C}} := [\hat{\mathbf{c}}_i]_{i=1}^{L} \in \mathbb{R}^{L \times 3}, \quad \hat{\mathbf{c}}_i = \frac{\mathbf{c}_i - \boldsymbol{\alpha}_t}{\boldsymbol{\beta}_t} \odot \boldsymbol{\beta}_{50} + \boldsymbol{\alpha}_{50}$$

Each normalized coordinate $\hat{\mathbf{c}}_t$ is mapped to $(h, s, l)$ using the function $D$. The results are arranged in a grid to produce a patch-level visualization $O_t$ of color at timestep $t$.

### 4.3. Intuition for Color Intervention

Although understanding how the model represents and processes color could enable color manipulation, how and when to intervene remains unclear. FM traverses from the noise to image distribution, with each end of the process implying a fundamentally different way to manipulate color.

At late timesteps, patch colors are fixed, and interventions must preserve inter-patch relations while remaining closed on the LCS. Hence, we shift the mean of the patches to the target color in the HSL space. In the LCS, this translates to adjusting hue, saturation, and lightness via rotation, shrinkage, and shift along the black-white axis, respectively.

However, color is not yet a property of individual patches early on. LCS coordinates of patches form an unstructured cloud where variance reflects unresolved possibilities, not color differences. Shrinkage collapses variance, destroying diversity instead of yielding coherent color changes. The mean decodes near grey by construction, rendering rotation largely ineffective for altering hue. But as Figure 3 shows, the patch coordinates' mean captures color, so a uniform distribution shift should achieve the desired color change.

Since FM treats the trajectory as an interpolation between image and noise, we interpolate between the color interventions with the same proportions. Section 5 qualitatively examines these two strategies and their interpolation.

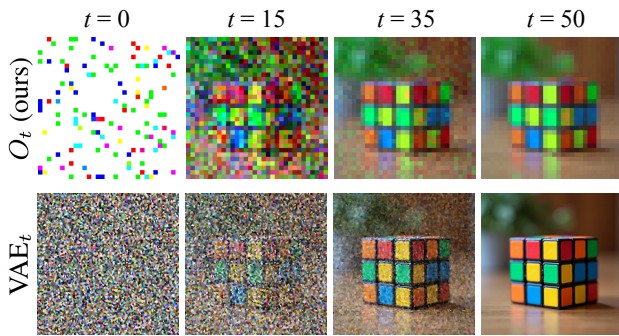

*Figure 5.* With our mid-generation color observation method (top), we validate our interpretation of the Latent Color Subspace (LCS) by predicting the final colors at intermediate timesteps. We compare these predictions with the VAE-decoded latents (bottom).

## 4.4. Concretizing Color Intervention

We consider a target color $\mathbf{y}^* = (h^*, s^*, l^*)$ in HSL format and its application at timestep $t$ to the LCS coordinates of image patches. Let $\mathbf{C} := [\mathbf{c}_i]_{i=0}^{L} \in \mathbb{R}^{L \times 3}$ denote the collection of $L$ patch coordinates at timestep $t$. To utilize the LCS–HSL approximation, we first normalize coordinates $\mathbf{C}$ to a reference timestep $t_{50}$, obtaining $\hat{\mathbf{C}} := [\hat{\mathbf{c}}_i]_{i=0}^{L} \in \mathbb{R}^{L \times 3}$. We now shift all patches to the desired color via the same intervention. Two types of interventions can achieve this.

**Type I: Direct LCS translation** We can compute the mean of the normalized coordinates $\bar{\mathbf{c}} = \frac{1}{L} \sum_{i=1}^{L} \hat{\mathbf{c}}_i$, encode the target color to LCS coordinates $\mathbf{c}^* = E(\mathbf{y}^*)$, and shift all patches by the same offset to get shifted coordinates

$$\hat{\mathbf{C}}' := [\hat{\mathbf{c}}_i']_{i=0}^{L}, \qquad \hat{\mathbf{c}}_i' = \hat{\mathbf{c}}_i + (\mathbf{c}^* - \bar{\mathbf{c}}).$$

**Type II: LCS shift via HSL space** Alternatively, we can decode $\hat{\mathbf{C}}$ to HSL colors:

$$\mathbf{Y} := [\mathbf{y}_i]_{i=0}^{L}, \qquad \mathbf{y}_i = D(\hat{\mathbf{c}}_i).$$

Then, obtain mean HSL color across patches, $\bar{\mathbf{y}} = \frac{1}{L} \sum_{i=1}^{L} \mathbf{y}_i$, and shift each patch in HSL space to produce the shifted HSL colors

$$\mathbf{Y}'' := [\mathbf{y}_i'']_{i=0}^{L}, \qquad \mathbf{y}_i'' = \mathbf{y}_i + (\mathbf{y}^* - \bar{\mathbf{y}}).$$

Encoding yields shifted LCS coordinates $\hat{\mathbf{C}}'' = E(\mathbf{Y}'')$.

**Type ⋆: Interpolation** We can also interpolate between both intervention types defining shifted LCS coordinates as

$$\hat{\mathbf{C}}^{\star} := \gamma_t \cdot \hat{\mathbf{C}}' + (1 - \gamma_t) \cdot \hat{\mathbf{C}}'',$$

where $\gamma$ is timestep-dependent interpolation coefficient derived from the FM scheduler. For all interventions, the resulting shifted LCS coordinates $\hat{\mathbf{C}}', \hat{\mathbf{C}}'', \hat{\mathbf{C}}^{\star}$ are denormalized back to timestep $t$, giving the final modified coordinates $\mathbf{C}', \mathbf{C}'', \mathbf{C}^{\star}$. We say we apply Type I/Type II/Type ⋆ intervention when we replace the original coordinates $\mathbf{C}$ with $\mathbf{C}'/\mathbf{C}''/\mathbf{C}^{\star}$. This process can be visualized in Figure 4.

*Table 1.* Perceptual color difference ($\Delta E_{00}$) between final images and latents at timestep $t$, observed by VAE-decoding ($\text{VAE}_t$) and interpreting the LCS ($O_t$). Note that by FLUX's design, $\text{VAE}_{50}$ is the final image and latents at $t = 0$ are pure noise.

*(a) $\Delta E_{00}$ computed per pixel*

| Dataset | Method | $t$ | | | | | |
| | | 0 | 10 | 20 | 30 | 40 | 50 |
|---|---|---|---|---|---|---|---|
| OBJECTS | $O_t$ (ours) | 44 | 31 | 20 | 14 | 12 | 12 |
| | $\text{VAE}_t$ | 26 | 21 | 15 | 9 | 4 | 0 |
| WALLS | $O_t$ (ours) | 45 | 23 | 11 | 8 | 7 | 7 |
| | $\text{VAE}_t$ | 33 | 25 | 19 | 12 | 5 | 0 |

*(b) $\Delta E_{00}$ computed for average pixel*

| Dataset | Method | $t$ | | | | | |
| | | 0 | 10 | 20 | 30 | 40 | 50 |
|---|---|---|---|---|---|---|---|
| OBJECTS | $O_t$ (ours) | 37 | 13 | 11 | 11 | 10 | 10 |
| | $\text{VAE}_t$ | 16 | 13 | 9 | 6 | 2 | 0 |
| WALLS | $O_t$ (ours) | 40 | 9 | 7 | 7 | 7 | 7 |
| | $\text{VAE}_t$ | 31 | 23 | 18 | 11 | 5 | 0 |

**Object-Localized Color Intervention** It is possible to alter the color of individual objects by supplying their corresponding patches in the latent space and applying the color intervention only to those regions. This approach simply requires masks, which can be obtained through a variety of methods depending on the available resources. For this work, we consider a conservative setting in which edits are applied mid-generation, without access to the final image and with only the target object specified. To obtain masks in this setting, we leverage segmentation maps derived from text–image cross-attention (Kim et al., 2025a), using activations from transformer layer 18. However, most mask-generation strategies for isolating target objects are compatible with our methodology.

## 5. Experiments

We measure perceptual color difference with HSL error ($\Delta H, \Delta S, \Delta L$), with $H$ in degrees and $S$ and $L$ as percentages, and CIEDE2000 (Luo et al., 2001) ($\Delta E_{00}$). Figure 6 presents Type I and Type II interventions; all other results use the Type ⋆ strategy at timestep 9. *Global* applies LCS color interventions to the entire image, whereas *local* applies them only to patches associated with the target object, with masks derived using Seg4Diff (Kim et al., 2025a) from activations of transformer layer 18.

### 5.1. Observation: Qualitative Evaluation

Figure 5 shows our observation method on the prompt "a photo of a rubik's cube on a table" across four timesteps. For comparison, we decode the corresponding latents with the VAE. Our method allows emerging colors to be clearly

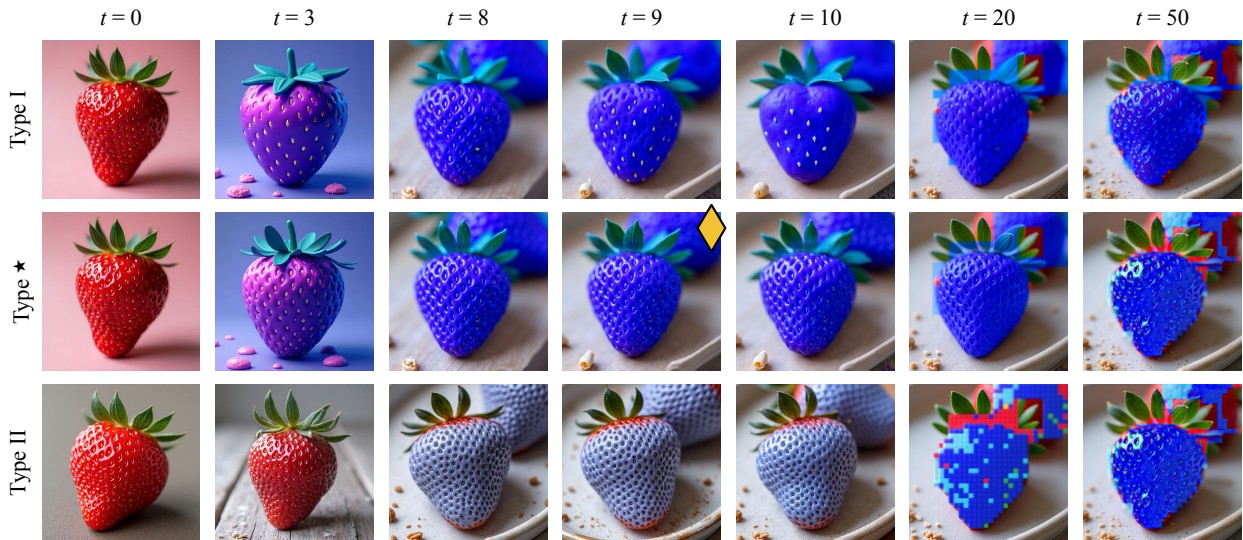

*Figure 6.* We investigate how to manipulate the LCS while preserving image quality. Color intervention shifting latent patches directly in LCS (Type I) disrupts texture, whereas shifting them via HSL (Type II) may have limited impact at early timesteps. Interpolating (Type ⋆) enables accurate color changes while preserving texture, so we select it at timestep 9 (♦) for the final method.

observed directly in latent space without decoding. Moreover, the accuracy with which we predict the final colors closely mirrors the fidelity of the decoded images, indicating the method reliably captures up-to-date information about the color dynamics occurring within the latent space at each timestep. More examples can be found in the Appendix.

## 5.2. Observation: Quantitative Evaluation

In Table 1, we evaluate how accurately our observation method represents the downscaled final image with $\Delta E_{00}$. We compare this to how well direct decoding of the latent with the VAE decoder represents the final image. We evaluate on two datasets: (i) GenEval's single-object task, scaling to more complex images (OBJECTS), and (ii) a dataset of 26 plain-colored walls described in Section 3 (WALLS).

At $t = 50$, our method achieves low color prediction errors of $\Delta E_{00} \leq 12$ on both datasets, for both per-pixel and averaged evaluations. In the per-pixel setting, errors fall to $\Delta E_{00} \leq 20$ as early as $t = 20$. But as expected, early timesteps are dominated by noise, resulting in less accurate predictions. In the averaged setting, performance is particularly strong: we obtain $\Delta E_{00} \leq 13$ on both datasets for all timesteps $t > 0$. Notably, for early timesteps, our method sometimes even outperforms direct VAE decoding. This suggests that our approach more effectively leverages the information encoded in global latent statistics, whereas the VAE decoder is trained only to decode the final latent representation. With our method, all of these quantities can be predicted directly in the latent space, without requiring the 50-million-parameter decoder to reconstruct the image.

## 5.3. Intervention: Qualitative Evaluation

As discussed in Section 4.4, Figure 6 considers three strategies for handling patches in latent space: Type I, Type II, and Type ⋆ (interpolation). We find Type ⋆ at timesteps $8 - 10$ most effective. In Type I, interventions can lose texture if applied too late (see $t = 10$), likely from unintended saturation changes; at very late timesteps, color fails to integrate and instead appears as a thin surface layer (see $t = 50$). In contrast, Type II can have little influence on the final image when applied at early timesteps (see $t = 3$). More generally, reliance on the model's internal, attention-based segmentation limits the feasibility of very early interventions (see $t = 3$ Type I), while the need for subsequent model "cleaning" to remove artifacts (see $t = 20$ Type II) and to smooth sharp, patch-induced boundaries (see $t = 50$) makes late interventions undesirable. Our proposed Type ⋆ approach addresses these limitations. In the critical timestep range effective for modifications (see $t = 8$–$10$), interpolation enables color integration while preserving more fine-grained texture than either Type I or II alone.

Figure 7 showcases our Type ⋆ color-intervention method on four prompts ("a photo of a teddy bear," "a photo of a flower," "a photo of a parrot", "a photo of a crystal ball on a table in mysterious library, it releases smoke") and six colors. Our method accurately identifies and manipulates individual objects' color while preserving overall structure. When applied to multi-color objects (see parrot), our method modifies the object so significant portions adopt the target color while remaining multi-colored overall. As the crystal ball illustrates, our interventions support more complex prompts and the base model can still adapt challenging light-

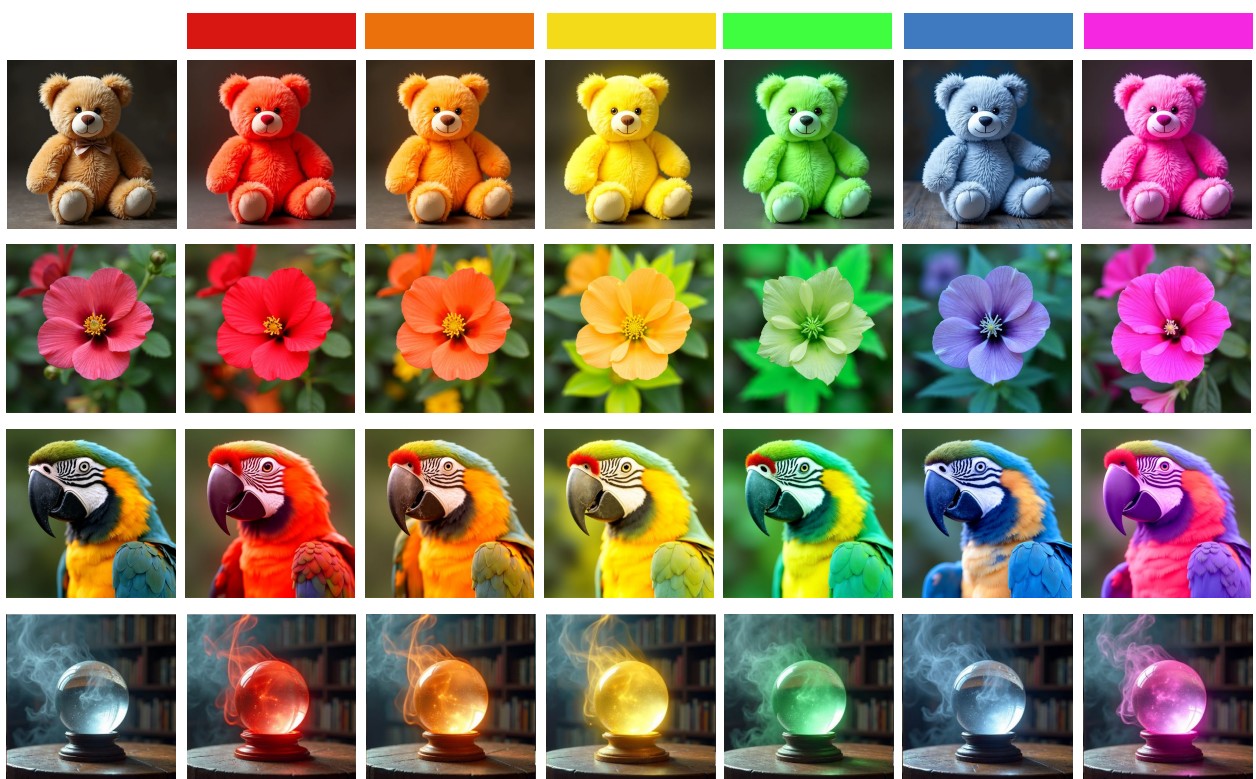

*Figure 7.* With our latent-space color interpretation, we can accurately guide objects toward target colors (top) while preserving much of the original image's high-level structure (left). Even multi-colored objects retain color diversity while shifting toward the target (row 3).

*Table 2.* Accuracy of our color intervention with GENEVAL's color task and the PRECISE tasks, including natural / plain images in 51 colors. We measure CIEDE2000 ($\Delta E_{00}$) and average distances in hue ($\Delta H$), saturation ($\Delta S$), and lightness ($\Delta L$) from target color. As baseline, we include results without specifying colors in the prompts (None). Our method effectively alters colors, affecting either entire image (global) or target object (local), without modifying this prompt. For comparison, we include color injected via prompt (Prompt).

| Color Injection | GENEVAL Acc ($\uparrow$) | PRECISE (NATURAL) $\Delta E_{00}$ ($\downarrow$) | $\Delta H$ ($\downarrow$) | $\Delta S$ ($\downarrow$) | $\Delta L$ ($\downarrow$) | PRECISE (PLAIN) $\Delta E_{00}$ ($\downarrow$) | $\Delta H$ ($\downarrow$) | $\Delta S$ ($\downarrow$) | $\Delta L$($\downarrow$) |
|---|---|---|---|---|---|---|---|---|---|
| None | 9% | 42 | 86° | 59% | 21% | 39 | 88° | 50% | 22% |
| Prompt | 79% | 23 | 30° | 43% | 14% | 20 | 24° | 36% | 13% |
| LCS ($\star$, local) | 68% | 14 | 14° | 37% | 6% | - | - | - | - |
| LCS ($\star$, global) | 72% | 16 | 18° | 33% | 9% | 11 | 8° | 22% | 8% |

ing, like reflections, to maintain coherence (e.g., reflection on table and smoke also shift to target color). Appendix B showcases our method's ability to generate fine-grained, novel hues and to control saturation and lightness. We refer the reader to the Appendix for a more detailed discussion of fine-grained preservation (Section H) and examples of multiple LCS edits to the same image (Section C).

## 5.4. Intervention: Quantitative Evaluation

Table 2 evaluates our intervention. As a baseline, we use FLUX with prompts specifying only objects, without color (*None*). Without modifying the prompt, our method injects color in two settings: global changes affecting the entire image and local changes applied only to the target object.

We compare to prompt-based color injection. We report accuracy on GenEval's *color* task (Ghosh et al., 2023) (see Appendix). Precise color control is not yet a well-established task, so existing benchmarks are limited; for more precise color measurements, we use 4,080 natural-images with 20 GenEval objects, 51 HSL colors, and 4 seeds (PRECISE (NATURAL), see Appendix). We isolate the object with masks from GenEval's object detector and compare the average masked color to the target color. For simpler images, we use 10 prompt-seed pairs that get plain images (PRECISE (PLAIN), see Appendix), without segmentation.

Mechanistic control alone raises the color accuracy from 9% to 72% on GenEval without changing the prompt, and approaches the 79% achievable with color-explicit prompts. Local changes achieve 68%, showing minimal error from

*Table 3.* Measured against the base prompt, we preserve original image structure more faithfully than prompt-based color changes.

| Color Inj. | IOU (↑) | SSIM (↑) | LPIPS (↓) | DINOv2 (↓) |
|---|---|---|---|---|
| Prompt | 0.60 | 0.46 | 0.49 | 0.36 |
| LCS (⋆, local) | 0.78 | 0.59 | 0.35 | 0.29 |
| LCS (⋆, global) | 0.88 | 0.56 | 0.36 | 0.23 |

*Table 4.* We compare our local method against other color generation techniques on PRECISE (NATURAL, SMALL), finding that it achieves the best color matching ($\Delta E_{00}$) while also better preserving structural consistency (SSIM, LPIPS).

| Color Injection | $\Delta E_{00}$ (↓) | SSIM (↑) | LPIPS (↓) |
|---|---|---|---|
| Prompt | 27 | 0.48 | 0.48 |
| BoN ($N = 50$) | 19 | 0.32 | 0.57 |
| ColorPeel (Butt et al., 2024) | 29 | 0.22 | 0.66 |
| ReNO (Eyring et al., 2024) | 23 | 0.25 | 0.61 |
| IP-Adapter (Ye et al., 2023) | 40 | 0.29 | 0.52 |
| LCS (⋆, local) | **10** | **0.62** | **0.32** |

segmentation masks. In PRECISE, our method has very accurate color control on plain images, with $\Delta E_{00} = 11$, $\Delta H = 8°$, and $\Delta L = 8\%$, compared to prompt-only results of $\Delta E_{00} = 20$, $\Delta H = 24°$, and $\Delta L = 13\%$. Even on complex images with local masks, accuracy remains high, with $\Delta E_{00} = 14$ and $\Delta H = 14°$. Overall, our approach achieves color precision beyond what prompting alone can provide, especially in hue.

### 5.5. Intervention Impact on Image Structure

On GenEval's color task, we examine how much our method alters image structure relative to the base generation, comparing it to prompt-based color changes with three similarity metrics: SSIM (Wang et al., 2004), LPIPS (Zhang et al., 2018) and distance in DINOv2 feature space (Oquab et al., 2024), all applied in grayscale to ignore the color. We also measure IoU between object masks from GenEval's detector. On all four metrics, our method more closely preserves the original image structure than modifying color via prompt. See Appendix for qualitative comparison.

### 5.6. Comparison to Other Color Techniques

To better understand the accuracy of our color interventions, we compare them against other methods for precise color generation. Accordingly, we compare our intervention against five alternative methods for color control: prompting, best-of-$N$ (BoN) sampling, ColorPeel (Butt et al., 2024), ReNO (Eyring et al., 2024), and IP-Adapter (Ye et al., 2023). Since many of these methods are significantly more computationally expensive than our interventions, we evaluate them on a reduced version of PRECISE (NATURAL), consisting of the same objects but a smaller subset of 15 colors, with one seed per prompt, for a total of 300 images.

In Table 4, our local LCS interventions not only achieve better color matching ($\Delta E_{00}$), but also preserve structure better (SSIM, LPIPS). Also worth noting is that LCS interventions are computationally inexpensive compared to other methods, often requiring additional per-color training (ColorPeel) or incurring higher inference-time costs (BoN, ReNO). Moreover, ours is the only color-control approach that improves interpretability of the underlying model. We refer the reader to Appendix Section J for a more comprehensive evaluation of color control, including details on the dataset and comparative methods, and to Section I for qualitative comparisons of structure preservation.

## 6. Conclusion

We find that color is represented in the VAE latent space of FLUX as an HSL-like bicone. We show that the corresponding latent directions can be used to both observe and intervene upon the generative process. We propose a fully training-free color-intervention method that enables control through purely mechanistic latent-space manipulation.

## Acknowledgments

This work was partially funded by the ERC (853489 - DEXIM) and the Alfried Krupp von Bohlen und Halbach Foundation, which we thank for their generous support. We are also grateful for partial support from the Pioneer Centre for AI, DNRF grant number P1. Mateusz Pach would like to thank the European Laboratory for Learning and Intelligent Systems (ELLIS) PhD program for support. The authors gratefully acknowledge the scientific support and resources of the AI service infrastructure LRZ AI Systems provided by the Leibniz Supercomputing Centre (LRZ) of the Bavarian Academy of Sciences and Humanities (BAdW), funded by Bayerisches Staatsministerium fur Wissenschaft und Kunst (StMWK).

## Impact Statement

This paper aims to advance the field of machine learning. While precise control over text-to-image generation models could, in principle, be misused for adversarial purposes, we do not identify any specific harmful applications from the ability to control color in text-to-image generative models.

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

## A. Colors in Timestep Experiments

The following colors are used in the timestep experiments, along with the average HEX value of the color:

Bright red: #D81511
Light red: #E7A0AD
Dark red: #78262F
Bright orange: #EA710B
Light orange: #F3C09C
Dark orange: #AA552F
Bright yellow: #F3DB1B
Light yellow: #ECD25B
Dark yellow: #D69613
Bright green: #26C812
Light green: #8DCF7A
Dark green: #1D4B32
Bright blue: #0FB3DF
Light blue: #94D3E3
Dark blue: #184166
Bright purple: #9360B4
Light purple: #CDB5E4
Dark purple: #59334C
Bright grey: #A3A4A3
Light grey: #BCBFBE
Dark grey: #3F4244
White: #E0E1E0
Black: #292929
Bright brown: #AA6B46
Light brown: #C8A171
Dark brown: #563727

# B. Additional Qualitative Results

We further illustrate the flexibility of our interpolated intervention method through additional qualitative examples. Figure 8 demonstrates fine-grained control over predicted hues, spanning a continuous range from red to magenta (#E60000, #E6002E, #E6005C, #E6008A, #E600B8, #E600E6). Figure 9 showcases saturation control by interpolating from blue to grey (#0000CC, #1A1AE6, #3333CC, #4D4DB3, #666699, #808080). Finally, Figure 12 highlights control over lightness, ranging from white to black through red (#DDDDDD, #F2B6B6, #D81612, #990000, #330000, #222222).

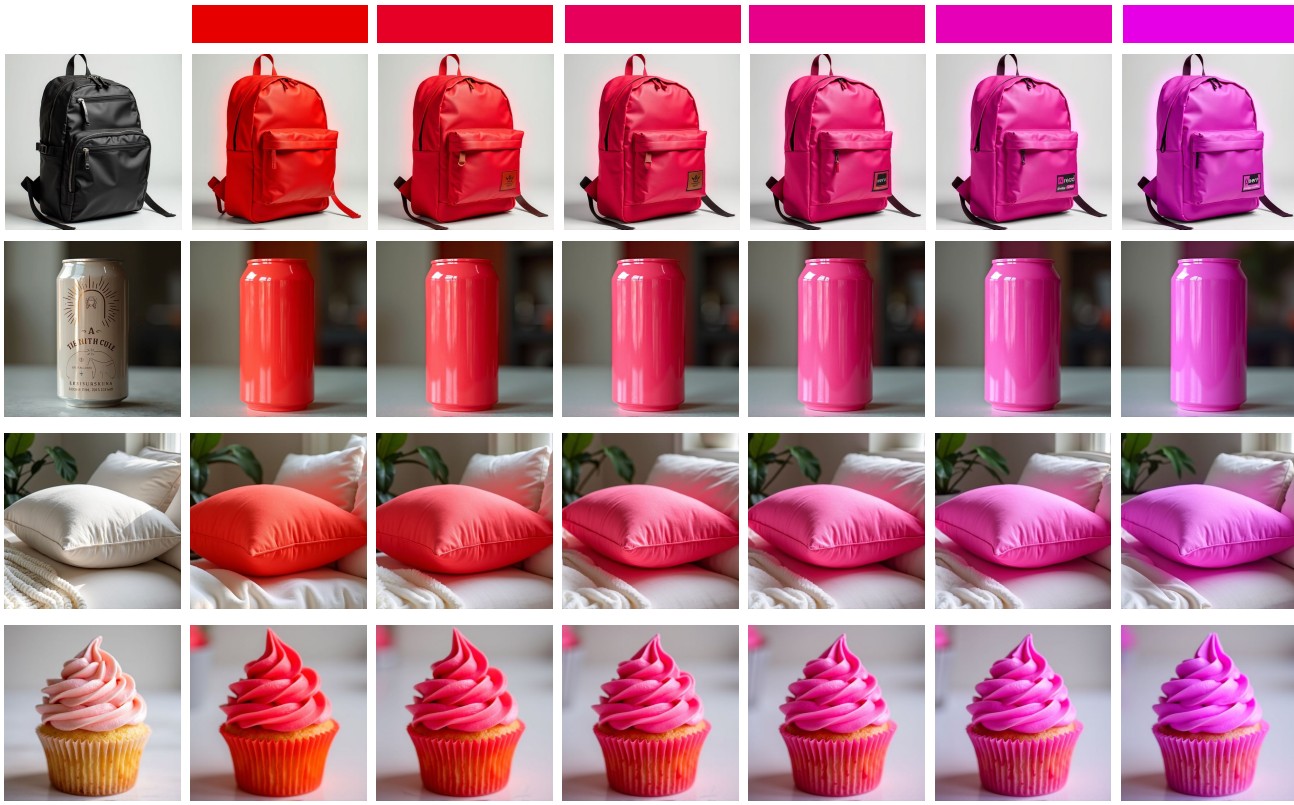

*Figure 8.* Demonstration of our interpolation method's ability to generate novel hues, spanning red to magenta.

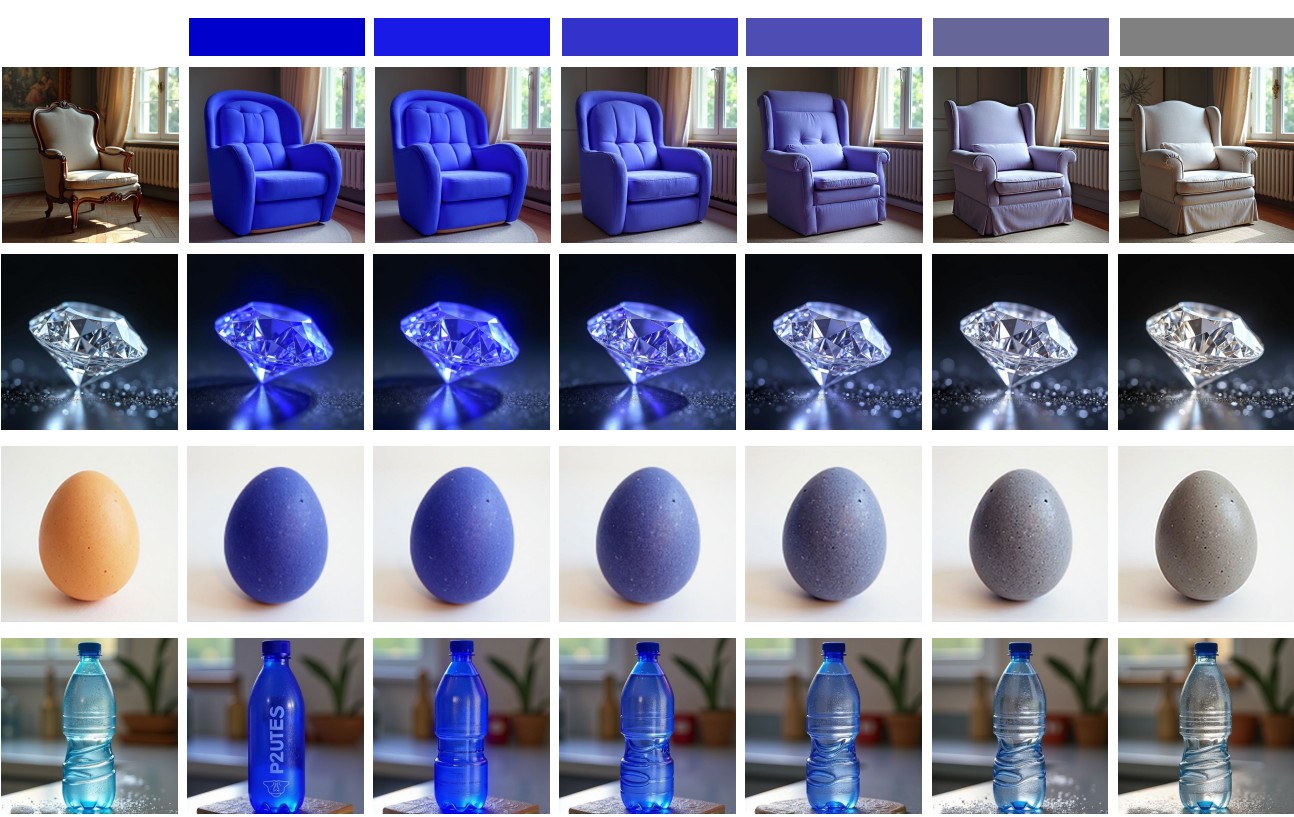

*Figure 9.* Demonstration of our interpolation method's ability to control saturation, spanning blue to grey.

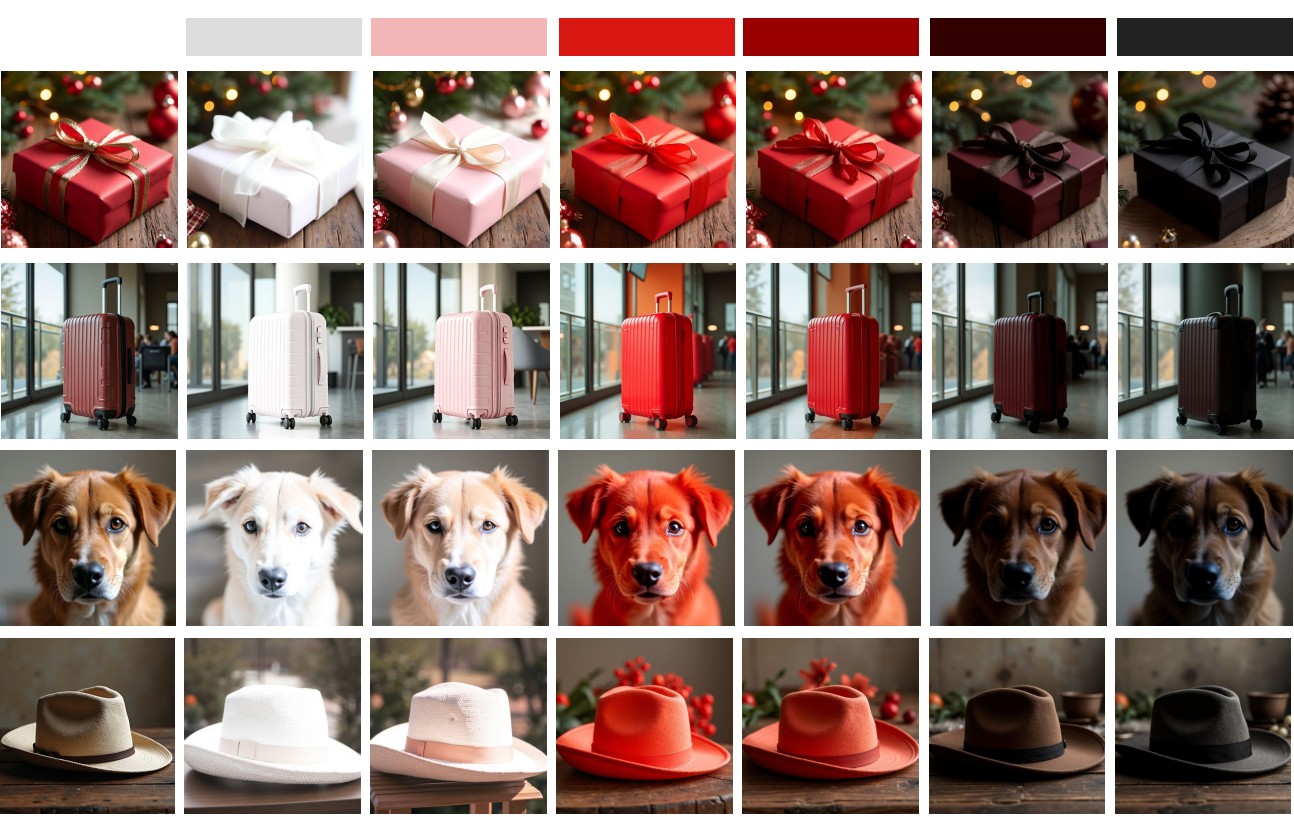

*Figure 10.* Demonstration of our interpolation method's ability to control lightness, spanning from white to black through red.

## C. Multiple Modifications

We include qualitative results suggesting that multiple color interventions within the LCS can be applied to a single image. In Figure 11, we present four examples demonstrating that two distinct interventions can simultaneously modify separate target objects to different colors. Notably, this remains feasible even when the objects are in close proximity (e.g., the dog and cat) or partially overlapping (e.g., the present behind the shoe).

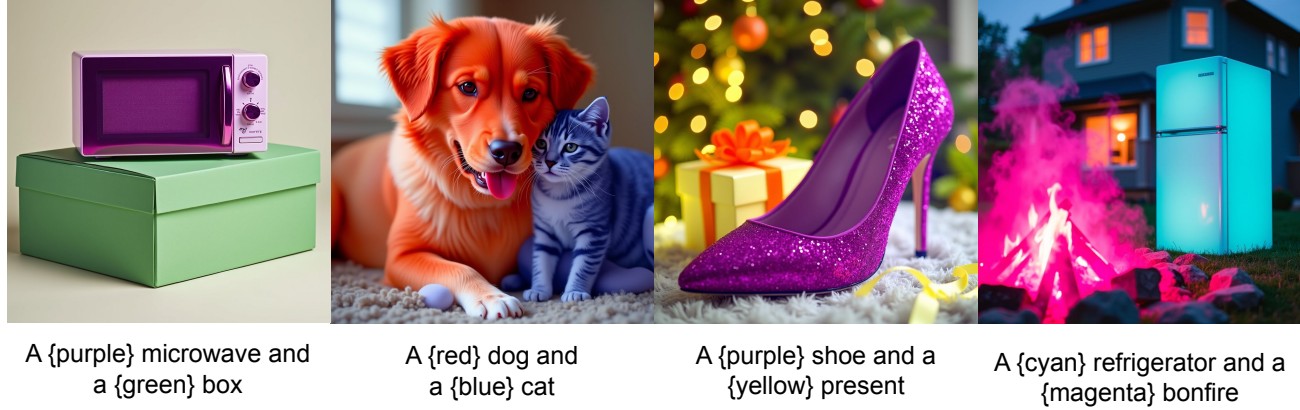

A {purple} microwave and a {green} box

A {red} dog and a {blue} cat

A {purple} shoe and a {yellow} present

A {cyan} refrigerator and a {magenta} bonfire

*Figure 11.* Examples demonstrating multiple LCS color interventions applied within a single image.

## D. PRECISE (OBJECTS) Settings

We include 20 objects from GenEval: "a photo of a frisbee", "a photo of a cow", "a photo of a broccoli", "a photo of a scissors", "a photo of a carrot", "a photo of a suitcase", "a photo of a elephant", "a photo of a cake", "a photo of a refrigerator", "a photo of a teddy bear", "a photo of a microwave", "a photo of a sheep", "a photo of a dog", "a photo of a zebra", "a photo of a bird", "a photo of a backpack", "a photo of a skateboard", "a photo of a banana", "a photo of a bear", "a photo of a fire hydrant".

We include a total of 51 colors, with 12 evenly distributed hues, with 4 types applied to each: light, dark, muted (unsaturated), and normal (saturated). The final three colors are black, grey, and white. Colors use following HEX codes and prompt names: 'Red': #FF0000, 'Orange': #FF7F00, 'Yellow': #FFFF00, 'Chartreuse': #7FFF00, 'Green': #00FF00, 'Spring Green': #00FF7F, 'Cyan': #00FFFF, 'Azure': #007FFF, 'Blue': #0000FF, 'Violet': #7F00FF, 'Magenta': #FF00FF, 'Rose': #FF007F, 'Dark Red': #7F0000, 'Dark Orange': #7F3F00, 'Dark Yellow': #7F7F00, 'Dark Chartreuse': #3F7F00, 'Dark Green': #007F00, 'Dark Spring Green': #007F3F, 'Dark Cyan': #007F7F, 'Dark Azure': #003F7F, 'Dark Blue': #00007F, 'Dark Violet': #3F007F, 'Dark Magenta': #7F007F, 'Dark Rose': #7F003F, 'Light Red': #FF7F7F, 'Light Orange': #FFBF7F, 'Light Yellow': #FFFF7F, 'Light Chartreuse': #BFFF7F, 'Light Green': #7FFF7F, 'Light Spring Green': #7FFFBF, 'Light Cyan': #7FFFFF, 'Light Azure': #7FBFFF, 'Light Blue': #7F7FFF, 'Light Violet': #BF7FFF, 'Light Magenta': #FF7FFF, 'Light Rose': #FF7FBF, 'Muted Red': #BF4040, 'Muted Orange': #BF7F40, 'Muted Yellow': #BFBF40, 'Muted Chartreuse': #7FBF40, 'Muted Green': #40BF40, 'Muted Spring Green': #40BF7F, 'Muted Cyan': #40BFBF, 'Muted Azure': #407FBF, 'Muted Blue': #4040BF, 'Muted Violet': #7F40BF, 'Muted Magenta': #BF40BF, 'Muted Rose': #BF407F, 'Black': #000000, 'White': #FFFFFF, 'Gray': #808080

Masks are taken from the object detector used for GenEval evaluation.

## E. PRECISE (PLAIN) Settings

In this setting, colors remain the same from the OBJECTS setting. Prompt-seed pairings are: ("a close-up photo of a wall", 12), ("a close-up photo of a paper sheet", 8), ("a photo of a clear sky", 4), ("a close-up photo of a plain sweater", 15), ("a close-up photo of a concrete floor", 5), ("a closeup of a plain rug", 3), ("a photo of a clear sky at night", 6), ("a close-up photo of sand", 0), ("a close-up photo of metal texture", 8), ("a close-up photo of wooden texture", 9)

Seeds were selected for uniform images.

# F. Additional Qualitative Observation Results

We include additional timesteps, to show our observation method in more detail on the prompt from the main paper, "a photo of a rubik's cube on the table. We show two additional examples as well, "a photo of a christmas tree" and "a photo of a fire truck".

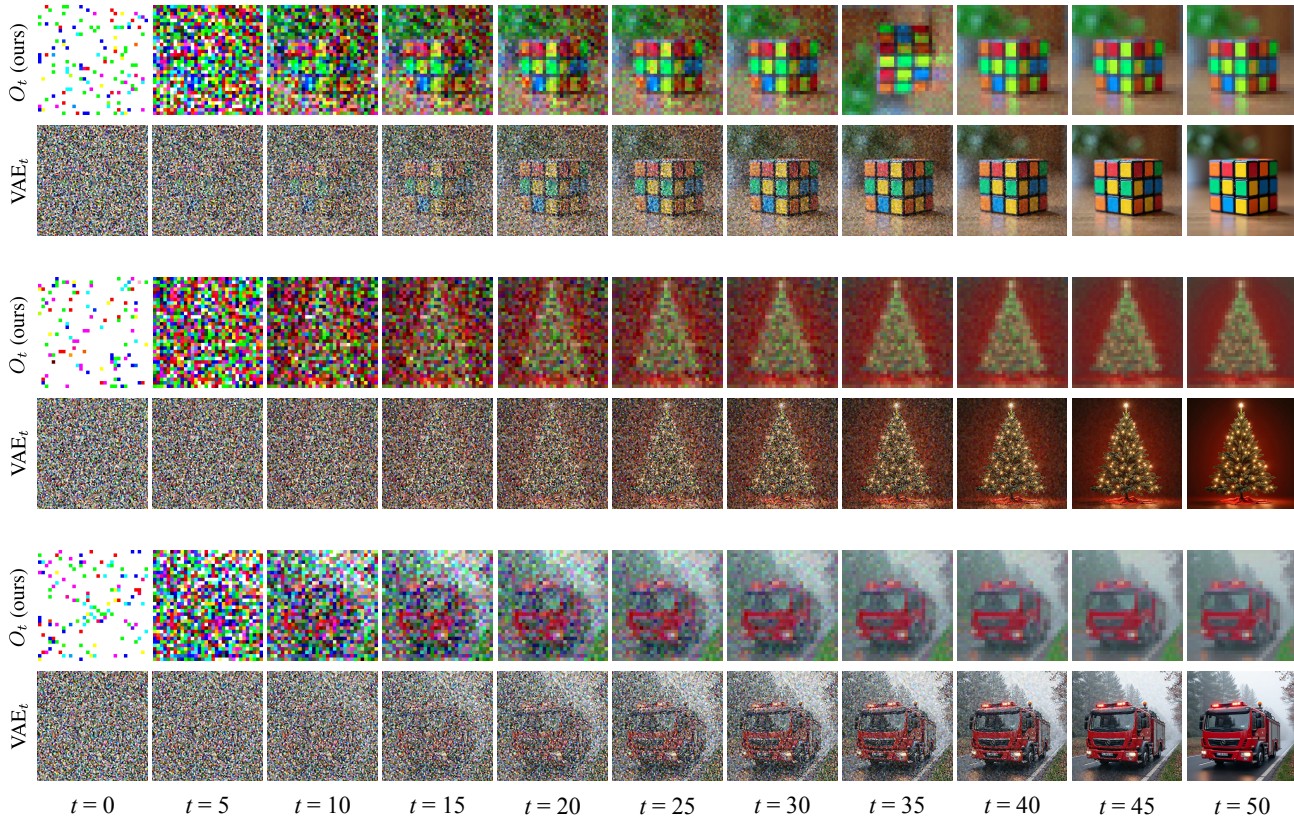

*Figure 12.* Qualitative examples of our observation method at a variety of timesteps, showcasing the accuracy of our Latent Color Space.

# G. Timestep Statistics of the Latent Color Subspace

We calculate the expected shift for each timesteps:

t = 0: 2.3413, -2.3586, 0.4266
t = 1: 2.3574, -2.3833, 0.4644
t = 2: 2.3638, -2.3904, 0.4883
t = 3: 2.3734, -2.3951, 0.5122
t = 4: 2.3831, -2.3993, 0.5384
t = 5: 2.3925, -2.4026, 0.5647
t = 6: 2.4023, -2.4047, 0.5919
t = 7: 2.4124, -2.4060, 0.6198
t = 8: 2.4226, -2.4064, 0.6484
t = 9: 2.4330, -2.4060, 0.6772
t = 10: 2.4437, -2.4051, 0.7065
t = 11: 2.4546, -2.4035, 0.7367
t = 12: 2.4659, -2.4011, 0.7668
t = 13: 2.4775, -2.3981, 0.7974
t = 14: 2.4897, -2.4009, 0.8312
t = 15: 2.5021, -2.4036, 0.8656
t = 16: 2.5148, -2.4065, 0.9008
t = 17: 2.5277, -2.4093, 0.9364
t = 18: 2.5408, -2.4123, 0.9727
t = 19: 2.5542, -2.4154, 1.0099
t = 20: 2.5680, -2.4186, 1.0481
t = 21: 2.5820, -2.4218, 1.0868
t = 22: 2.5963, -2.4252, 1.1263
t = 23: 2.6110, -2.4288, 1.1672
t = 24: 2.6261, -2.4324, 1.2090
t = 25: 2.6416, -2.4363, 1.2520
t = 26: 2.6575, -2.4403, 1.2957
t = 27: 2.6738, -2.4444, 1.3406
t = 28: 2.6904, -2.4485, 1.3865
t = 29: 2.7074, -2.4529, 1.4336
t = 30: 2.7250, -2.4574, 1.4818
t = 31: 2.7432, -2.4621, 1.5314
t = 32: 2.7618, -2.4669, 1.5823
t = 33: 2.7810, -2.4720, 1.6344
t = 34: 2.8006, -2.4771, 1.6878
t = 35: 2.8209, -2.4826, 1.7430
t = 36: 2.8418, -2.4883, 1.7995
t = 37: 2.8631, -2.4944, 1.8578
t = 38: 2.8853, -2.5005, 1.9179
t = 39: 2.9080, -2.5066, 1.9793
t = 40: 2.9313, -2.5132, 2.0426
t = 41: 2.9555, -2.5199, 2.1082
t = 42: 2.9804, -2.5268, 2.1756
t = 43: 3.0060, -2.5338, 2.2450
t = 44: 3.0328, -2.5411, 2.3172
t = 45: 3.0603, -2.5486, 2.3914
t = 46: 3.0889, -2.5561, 2.4682
t = 47: 3.1189, -2.5640, 2.5482
t = 48: 3.1497, -2.5725, 2.6302
t = 49: 3.1824, -2.5796, 2.7175
t = 50: 3.2152, -2.5889, 2.8050

We provide the mean magnitudes after centering as well:

t = 0: 0.0163, 0.0172, 0.0295
t = 1: 0.0905, 0.0716, 0.0999
t = 2: 0.1345, 0.1123, 0.1544
t = 3: 0.1826, 0.1491, 0.2065
t = 4: 0.2360, 0.1899, 0.2630
t = 5: 0.2904, 0.2316, 0.3202
t = 6: 0.3471, 0.2749, 0.3793
t = 7: 0.4050, 0.3191, 0.4394
t = 8: 0.4640, 0.3641, 0.5003
t = 9: 0.5231, 0.4091, 0.5611
t = 10: 0.5834, 0.4547, 0.6228
t = 11: 0.6456, 0.5016, 0.6861
t = 12: 0.7077, 0.5481, 0.7488
t = 13: 0.7713, 0.5958, 0.8127
t = 14: 0.8410, 0.6496, 0.8866
t = 15: 0.9119, 0.7044, 0.9616
t = 16: 0.9845, 0.7605, 1.0386
t = 17: 1.0578, 0.8172, 1.1163
t = 18: 1.1325, 0.8750, 1.1957
t = 19: 1.2094, 0.9344, 1.2771
t = 20: 1.2880, 0.9953, 1.3606
t = 21: 1.3680, 1.0571, 1.4453
t = 22: 1.4498, 1.1205, 1.5321
t = 23: 1.5341, 1.1858, 1.6216
t = 24: 1.6206, 1.2526, 1.7131
t = 25: 1.7094, 1.3214, 1.8072
t = 26: 1.7998, 1.3913, 1.9030
t = 27: 1.8927, 1.4633, 2.0014
t = 28: 1.9879, 1.5370, 2.1022
t = 29: 2.0854, 1.6126, 2.2056
t = 30: 2.1853, 1.6900, 2.3114
t = 31: 2.2881, 1.7696, 2.4202
t = 32: 2.3939, 1.8515, 2.5321
t = 33: 2.5021, 1.9354, 2.6467
t = 34: 2.6133, 2.0215, 2.7642
t = 35: 2.7280, 2.1106, 2.8857
t = 36: 2.8455, 2.2017, 3.0101
t = 37: 2.9668, 2.2957, 3.1386
t = 38: 3.0921, 2.3929, 3.2712
t = 39: 3.2204, 2.4922, 3.4067
t = 40: 3.3523, 2.5946, 3.5464
t = 41: 3.4888, 2.7006, 3.6911
t = 42: 3.6292, 2.8097, 3.8398
t = 43: 3.7741, 2.9222, 3.9931
t = 44: 3.9247, 3.0394, 4.1527
t = 45: 4.0793, 3.1597, 4.3168
t = 46: 4.2393, 3.2843, 4.4866
t = 47: 4.4053, 3.4142, 4.6636
t = 48: 4.5760, 3.5480, 4.8461
t = 49: 4.7541, 3.6886, 5.0383
t = 50: 4.9407, 3.8364, 5.2390

## H. Improving Fine-Grained Preservation

While our color intervention strategy in the LCS largely preserves non-color semantics (see Figure 6, $t = 50$, where only minimal non-color information is altered), the basic setting can still introduce slight variations in fine details, due to the modification being applied at earlier timesteps (see Figure 6, $t = 0$), where the intervention tends to produce greater overall variation. For some use cases, this behavior may even be desirable, as it can steer the semantics toward greater realism—for instance, making the red steak appear raw or the green stump appear moss-covered (see Figure 13).

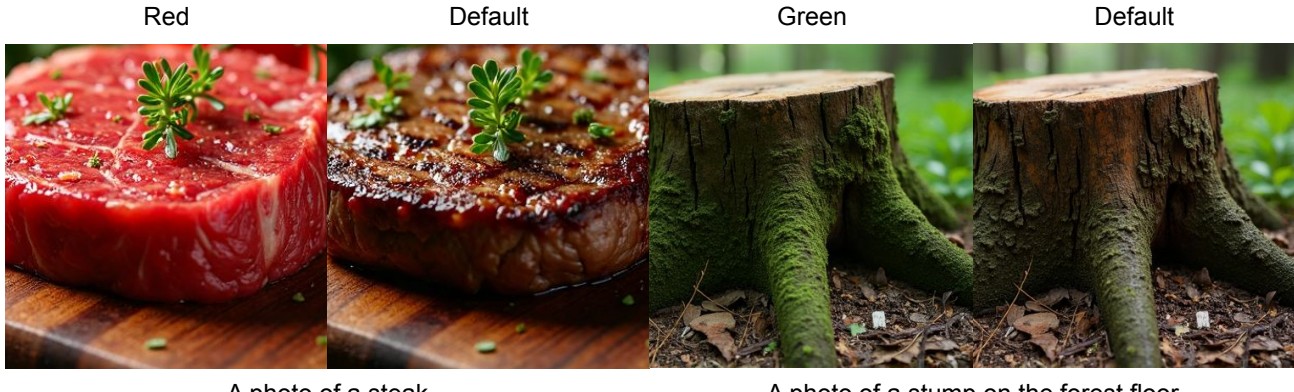

*Figure 13.* Fine-grained modifications can be beneficial, as they may update the objects in ways that make them more realistic.

However, for other use cases—such as image editing—these changes may be undesirable. Since they arise primarily from early modifications that alter the latent trajectory, they can be mitigated by applying the intervention at later timesteps (e.g., $t = 20–25$), where the trajectory is more stable. The only additional requirement is cleaner bounding boxes. In Figure 14, we compare the basic setting from the main paper using the prompt "a photo of a shoe," where broad semantics and structural information are preserved but fine-grained details may vary, against a setup with improved bounding boxes and later timestep intervention, where fine-grained details are substantially better preserved.

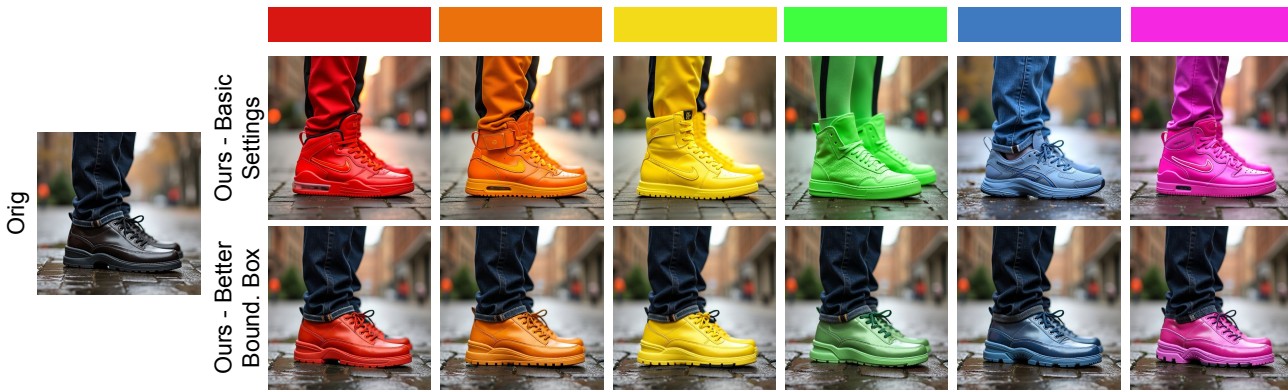

*Figure 14.* Color interventions in the basic setting preserve structural and high-level semantic information, but with better bounding boxes and a later intervention timestep, even fine-grained details can be preserved.

# I. Structure Preservation Qualitative Comparison

We compare the structural impact of color changes induced by the LCS with other color-generation methods in Figure 15: prompt-based, Best of $N = 50$ (BoN, where 50 images are generated and the best color-matching seed is selected), ColorPeel (Butt et al., 2024), ReNO (Eyring et al., 2024), and IPAdapter (Ye et al., 2023). We find that LCS-based intervention more accurately preserves both structural and fine-grained details than the alternatives. While IP-Adapter (Ye et al., 2023) performs second best, LCS still better maintains background consistency and fine-grained details.

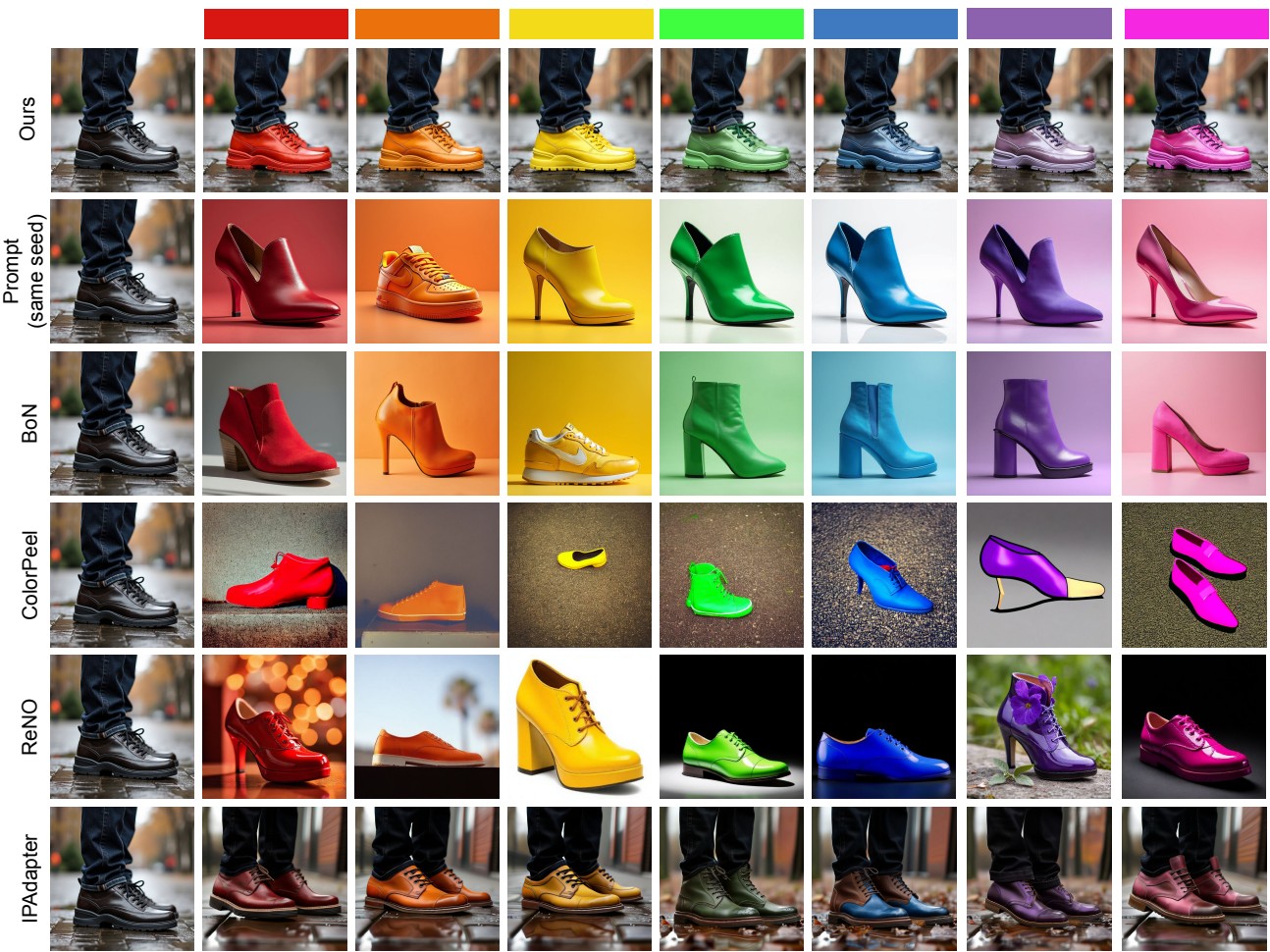

*Figure 15.* Comparison of the structural changes in images when comparing intervention via the LCS to those by other color generation methods.

*Table 5.* Comparison to other color-control alternatives.

| Color Inj. | PRECISE (NATURAL, SMALL) | | | |
|---|---|---|---|---|
| | $\Delta E_{00}$ ($\downarrow$) | $\Delta H$ ($\downarrow$) | $\Delta S$ ($\downarrow$) | $\Delta L$ ($\downarrow$) |
| None | 45 | 80 | 62 | 19 |
| Prompt | 27 | 38 | 42 | 14 |
| Best of $N = 10_{None}$ | 39 | 72 | 64 | 14 |
| Best of $N = 20_{None}$ | 37 | 70 | 63 | 15 |
| Best of $N = 50_{None}$ | 34 | 68 | 64 | 14 |
| Best of $N = 10_{Prompt}$ | 22 | 34 | 43 | 10 |
| Best of $N = 20_{Prompt}$ | 20 | 32 | 43 | 9 |
| Best of $N = 50_{Prompt}$ | 19 | 32 | 41 | 9 |
| Color Peel (Butt et al., 2024) | 29 | 54 | 43 | 12 |
| $ReNO_{SDXL}$ (Eyring et al., 2024) | 24 | 30 | 46 | 11 |
| $ReNO_{FLUX}$ (Eyring et al., 2024) | 23 | 34 | 38 | 12 |
| LCS($\star$, local) | **10** | **12** | 25 | **7** |
| LCS($\star$, global) | 11 | 16 | **22** | 8 |

## J. Comparison with Other Color Control Methods

Precise color control is a challenging task which cannot be solved easily training free. Specifically, most existing methods require high compute, either in the form of training or at inference time. This makes it impossible to scale generation to the 4,080 images used in the natural setting in the main paper (20 objects, 51 colors, 4 seeds). Instead, we use a subset including the 15 most basic colors (Red, Orange, Yellow, Chartreuse, Green, Spring Green, Cyan, Azure, Blue, Violet, Magenta, Rose, Black, White, Gray), the same 20 objects, and 1 seed for a total of 300 tested images.

The first additional comparison method is *best of N*, in which $N$ images are generated, and the image with the lowest $\Delta E_{00}$ among them is selected. This results in $N$ times the inference cost, which can become extreme for high $N$. We include versions of this baseline applied to both our $None$ (prompt given without color) and $Prompt$ (color specified in prompt), and test $N = 10, 20, 50$.

The next comparison method is $ColorPeel$ (Butt et al., 2024), a training-based method that requires optimizing parameters for each target color. Therefore, computational costs scale by the number of unique colors required, unlike our intervention method which does not incur additional costs for new colors.

Finally, we compare with $ReNO$ (Eyring et al., 2024), which can be used for more accurate prompt following. ReNO leverages test-time noise optimization to maximize prompt-following, therefore incurring per-image optimization cost.

Despite incurring less cost than any of these methods, our intervention achieves more precise color match in terms of $\Delta E_{00}$, $\Delta H$, $\Delta S$, and $\Delta L$. It is also the only method among these that leverages insights of inner model workings to improve the capabilities of the base model, rather than adding more uninterpretable optimization.

*Table 6.* Our intervention method's performance on high (bright) and low (muted) saturation colors.

| Color Inj. | PRECISE (NATURAL, BRIGHT) | | | | PRECISE (NATURAL, MUTED) | | | |
|---|---|---|---|---|---|---|---|---|
| | $\Delta E_{00}$ ($\downarrow$) | $\Delta H$ ($\downarrow$) | $\Delta S$ ($\downarrow$) | $\Delta L$ ($\downarrow$) | $\Delta E_{00}$ ($\downarrow$) | $\Delta H$ ($\downarrow$) | $\Delta S$ ($\downarrow$) | $\Delta L$ ($\downarrow$) |
| None | 47 | 90 | 73 | 15 | 39 | 90 | 27 | 15 |
| Prompt | 29 | 28 | 51 | 10 | 21 | 28 | 21 | 10 |
| LCS($\star$, local) | 11 | 8 | 28 | 7 | 14 | 16 | 19 | 6 |
| LCS($\star$, global) | 13 | 8 | 24 | 9 | 17 | 18 | 16 | 11 |

*Table 7.* Our intervention method's performance on high (light) and low (dark) lightness colors.

| Color Inj. | PRECISE (NATURAL, LIGHT) | | | | PRECISE (NATURAL, DARK) | | | |
|---|---|---|---|---|---|---|---|---|
| | $\Delta E_{00}$ ($\downarrow$) | $\Delta H$ ($\downarrow$) | $\Delta S$ ($\downarrow$) | $\Delta L$ ($\downarrow$) | $\Delta E_{00}$ ($\downarrow$) | $\Delta H$ ($\downarrow$) | $\Delta S$ ($\downarrow$) | $\Delta L$ ($\downarrow$) |
| None | 48 | 90 | 73 | 37 | 35 | 90 | 73 | 15 |
| Prompt | 26 | 30 | 54 | 25 | 17 | 22 | 53 | 7 |
| LCS($\star$, local) | 17 | 16 | 53 | 6 | 16 | 16 | 55 | 5 |
| LCS($\star$, global) | 17 | 20 | 52 | 8 | 18 | 18 | 45 | 8 |

## K. Further Quantitative Results for Color Interventions

We break down our main results into sub-categories, to develop a fine-grained understanding of our color intervention method's performance under different settings in Tables 6 and 7. *Bright* and *Muted* refer to high and low saturation colors, and *Light* and $Dark$ refer to high and low lightness.

We also include metrics in other color representation systems, including: $\Delta H_{HCL}$ (hue as defined in HCL), $\Delta C_{HCL}$ (chroma as defined in HCL), $\Delta L_{HCL}$ (luminance as defined in HCL), $\Delta a^*$ (green-red as defined in CIELAB), and $\Delta b^*$ (blue-yellow as defined in CIELAB). These metrics further confirm that LCS interventions enable accurate color control.

*Table 8.* Metrics from other color systems evaluated on PRECISE (NATURAL).

| Color Inj. | $\Delta H_{HCL}$ ($\downarrow$) | $\Delta C_{HCL}$ ($\downarrow$) | $\Delta L_{HCL}$ ($\downarrow$) | $\Delta a^*$ ($\downarrow$) | $\Delta b^*$ ($\downarrow$) |
|---|---|---|---|---|---|
| No Color | 0.83 | 0.47 | 0.27 | 0.44 | 0.46 |
| Prompt | 0.30 | 0.34 | 0.17 | 0.30 | 0.27 |
| LCS($\star$, local) | **0.16** | 0.30 | **0.09** | 0.23 | 0.21 |
| LCS($\star$, global) | 0.18 | **0.28** | 0.11 | **0.22** | **0.20** |

## L. LCS in other models

We verify the existance of an LCS in other models. Specifically, we confirm its presence in SD3.5 (Esser et al., 2024), Qwen-Image (Wu et al., 2025), and FLUX.2 (Black Forest Labs, 2025a) by repeating the PCA experiments in each model's VAE space. In all cases, the first three principal components explain 100% of the variance, and their structure closely resembles HSL when visualized (see Figure 16).

This provides preliminary evidence that the LCS is not specific to FLUX.1, but may instead generalize across architectures.

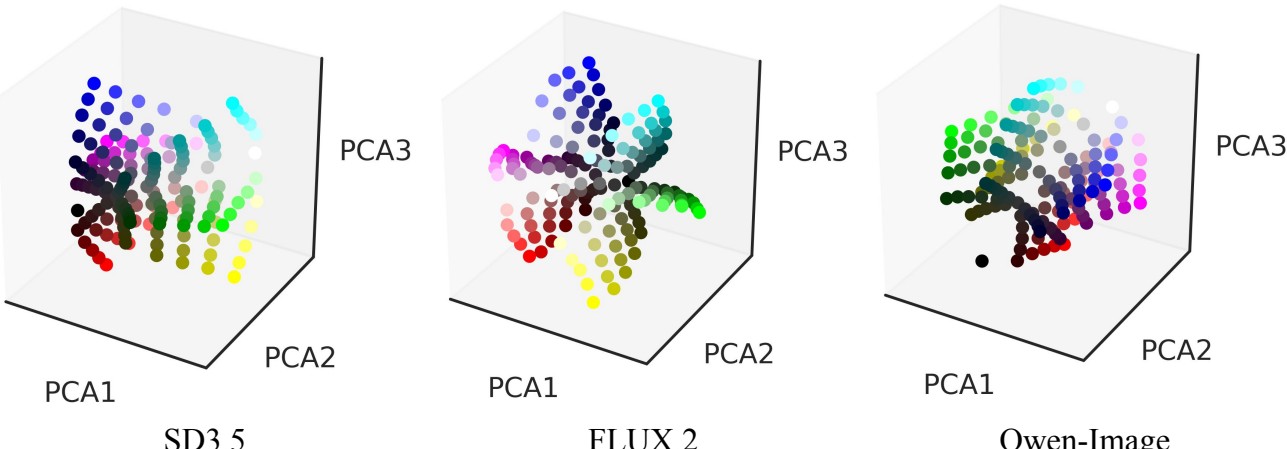

*Figure 16.* Visualizations of the LCS in SD3.5, Qwen-Image, and FLUX.2.

## M. Illustration of the LCS–HSL Bijection

We include an illustration of the bijection between the LCS and HSL spaces to provide additional intuition.

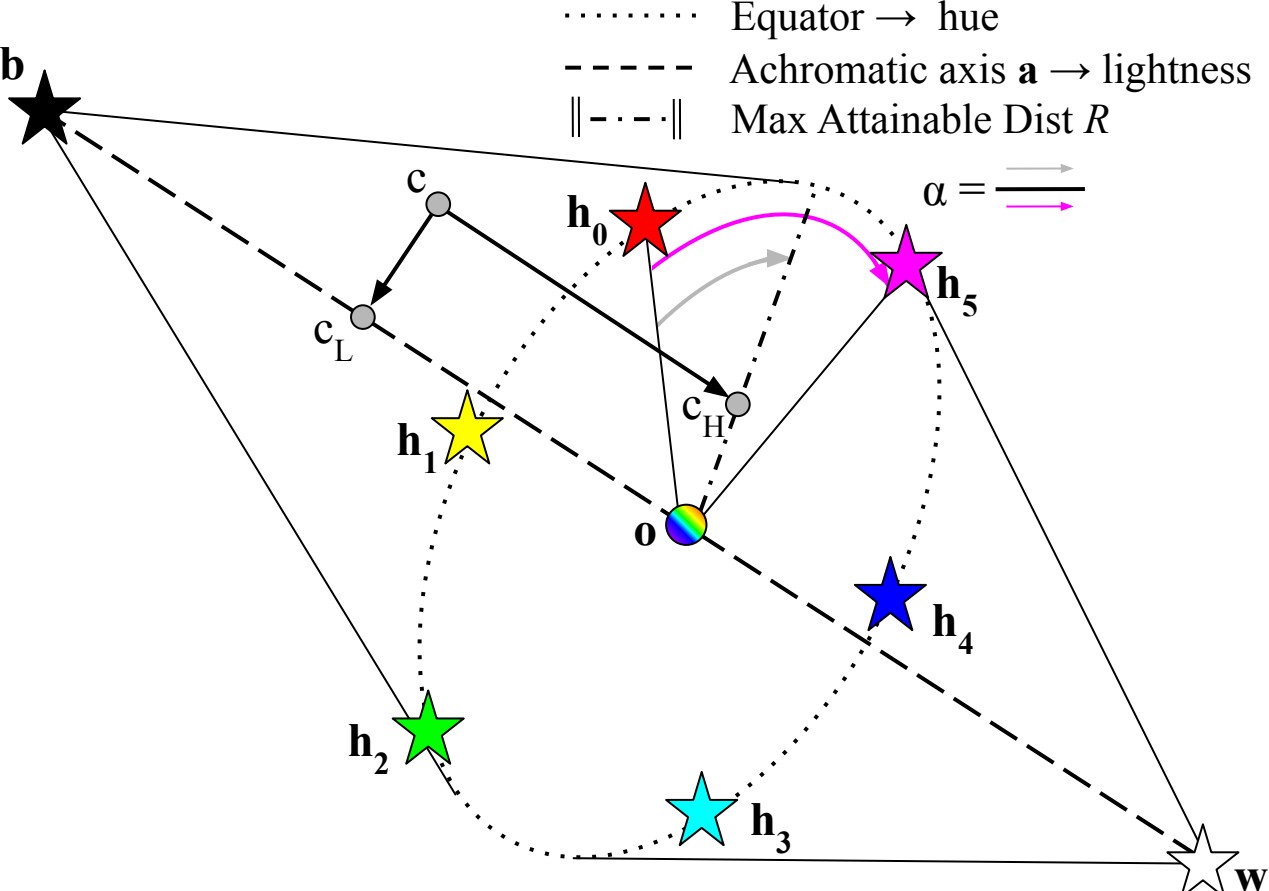

*Figure 17.* Illustration of the bi-jection between HSL and LCS space.

