# OpenReview forum: "The Latent Color Subspace: Emergent Order in High-Dimensional Chaos"
_ICML.cc/2026/Conference — ICML 2026 regular_

### Official Review · Reviewer_aBcj · 2026-03-11

**Soundness:** 3
**Presentation:** 1
**Significance:** 1
**Originality:** 3
**Overall Recommendation:** 3
**Confidence:** 3

**Summary:**

This paper studies how color is represented in the latent space of FLUX. The authors find that color information forms a small structured subspace aligned with hue, saturation, and lightness. They introduce methods to manipulate this latent color subspace during generation, allowing training-free control of image colors with little change of the image content.

**Compliance With Llm Reviewing Policy:**

Affirmed.

**Key Questions For Authors:**

What are the metrics: perceptual color difference and CIEDE2000

Can you compare your method with other training-required methods in terms of color intervention quality?

**Limitations:**

No limitation has been mentioned in the paper. See weakness for suggestions

**Strengths And Weaknesses:**

## Strength
1. The analysis and the finding of Latent Color Subspace in Flux-VAE are interesting

2. The proposed color intervention method is training-free.


## Weakness

1. The presentation complicates the whole paper, even though the method is easy to understand. The readability is low, especially for researcher do not often work with 'color'. Since the value of this paper is its practical application, I suggest the authors improve the writing (particularly Section 4 and Section 5) to fit more readers.

2. The VAE described in section 3.1 is not the VAE used in Latent Diffusion. The VAE of Latent Diffusion is trained normally with LPIPS loss and GAN loss, and with a very weak KL divergence (even some papers remove it when training the AE)

3. The proposed color intervention method **has visually changed the image semantics**, which hinders its application. For example, Row 2 of Figure 7 shows that the shoe type and jean type have been changed after the color intervention. Figure 6 also shows that the changed color has **obvious color artifacts** (a mixture of original color and target color around the object edge)

4. The paper only benchmarks on FLUX; do the observation and method also fit other latent diffusion models? (e.g. Stable Diffusion, Qwen-Image)?

5. Some reference errors:
* author names are wrong in 'Mantecon, H. L., Gomez-Villa, A., Qin, J., Butt, M. A.,Raducanu, B., Vazquez-Corral, J., van de Weijer, J., and Wang, K. Leveraging semantic attribute binding for free-lunch color control in diffusion models. WACV, 2026'. While the correct one is 'Laria, Héctor, et al. "Leveraging semantic attribute binding for free-lunch color control in diffusion models." WACV. 2026.'

* in Line 499, duplicate references:
Shum, K. C., Hua, B.-S., Nguyen, D. T., and Yeung, S.-K.Color alignment in diffusion. 2025a. Shum, K. C., Hua, B.-S., Nguyen, D. T., and Yeung, S.-K.Color alignment in diffusion. In CVPR, 2025b.

* in Line 540, duplicate references:
Zhang, L., Rao, A., and Agrawala, M. Adding conditional control to text-to-image diffusion models. ICCV, 2023a.
Zhang, L., Rao, A., and Agrawala, M. Adding conditional control to text-to-image diffusion models. In ICCV, 2023b

---

> ### Author Rebuttal · Authors · 2026-03-30
>
> We thank the Reviewer for recognizing our findings as interesting and our method as training-free. We will incorporate all suggested changes in the final version.
>
> **(W1) Intuitions for sections 4 and 5**
>
> We thank the Reviewer for the feedback on clarity. To improve readability, we will incorporate additional intuition into the text and complement much of the information with figures. This will provide a more intuitive illustration of how we map between HSL and the LCS, as well as how our interventions are applied.
>
>
>
> **(W2 / W5) FLUX VAE and references**
>
> Thank you for the corrections, we will revise the paper accordingly. Regarding the VAE, as the available FLUX training documentation is somewhat limited, we based this section on comments in the Hugging Face repository, but will refer to the FLUX.1 Kontext paper instead, which is aligned with the Reviewer’s suggestion. We note that this misunderstanding is limited to the description of the training process and does not affect the findings presented in the paper.
>
>
>
> **(W3) Correction of artifacts and fine-grained flexibility**
>
> It is important to note that Figure 6 presents an ablation study; only the middle row at t=9 corresponds to the final method (and it is free of such artifacts). As discussed in Section 5.3, the figure highlights the key challenges that constrain which timesteps are suitable for intervention and motivates our choice of an interpolation strategy. The final hyperparameter selection specifically addresses these artifacts, leading to the improved results shown in Figure 7.
>
> Regarding the observed changes in semantics, these primarily stem from the application of mid-generation interventions rather than modifications via the LCS; we refer the Reviewer to our response to Reviewer Mftg Q2 for further discussion. We also emphasize that the discovery of the LCS—an interpretable subspace within the VAE latent space—is itself a surprising result. As shown in Figure 6, how to translate the LCS into an effective intervention strategy is not obvious: it requires careful reasoning about how the FM model evolves distributions across timesteps, along with experimental validation. Together, these findings provide fundamental insight into the model’s internal organization and show that this structure is causal, enabling controlled generation and opening the door to a wide range of applications. While some use cases (e.g., image editing) demand fine-grained preservation, others (e.g., color-conditioned generation) may not. We hope this work motivates future research that adapts these insights to specific tasks—particularly those requiring stricter constraints—likely in combination with pre-existing task-specific strategies. We demonstrate that our initial intervention strategy preserves core structural and semantic information, while leaving the incorporation of task-specific constraints to future work.
>
> **(W4) LCS in other models**
>
> We verify that a LCS also exists in other models. Specifically, we confirm its presence in SD3.5, Qwen-Image, and FLUX.2 by repeating the PCA experiments in each model’s VAE space. In all cases, the first three principal components explain 100% of the variance, and their structure closely resembles HSL when visualized.
>
> We further conduct preliminary intervention experiments using 10 objects and 3 colors. These tests evaluate whether (1) the target object is preserved after modification—ensuring the prompt remains faithfully followed—and (2) the color intervention is successful. Across all three models, both criteria are satisfied with over 97% accuracy.
>
> Together, these results demonstrate that the VAEs in these models organize color within a highly interpretable three-dimensional subspace, and provide initial evidence that similar interventions can be applied effectively.
>
> **(Q1) Metric clarification**
>
> The perceptual color difference metrics are ΔH (change in hue, representing the shade or tint), ΔS (change in saturation, indicating how bright or muted a color is), ΔL (change in lightness, reflecting how light or dark a color appears), and CIEDE2000. CIEDE2000 (referred to in our work as ΔE) is a holistic color difference metric used in color science to measure distances between colors in CIELAB space, which is designed for perceptual uniformity. Unlike simpler metrics such as L2 distance in RGB, ΔE is proportional to perceived color differences, better reflecting human visual similarity.
>
>
>
> We will include intuitive explanations and the corresponding equations in the manuscript.
>
>
>
> **(Q2) Comparison with other methods**
>
> We refer the Reviewer to the response to Reviewer Z14G Q2. We evaluate our method against state-of-the-art color-intervention strategies on a shared benchmark, demonstrating that it is not only the only approach grounded in interpretable principles but also achieves the most precise color edits.

---

> > ### Author Rebuttal · Reviewer_aBcj · 2026-04-03
> >
> > Thanks for the response, but the explanation of 'Correction of artifacts and fine-grained flexibility' does not address my concern. Considering the significance and drawbacks of this paper, I maintain my rating.

---

> > > ### Author Response · Authors · 2026-04-08
> > >
> > > We are pleased to have provided clarity on many issues, appreciate your update on the concerns requiring further explanation, and are happy to address them more comprehensively.
> > >
> > > **Fine-grained preservation**
> > >
> > > Although outside the scope of the original task, we are excited for the opportunity to demonstrate that when desired even fine details can be well preserved by small adjustments like more accurate masks and later interventions. When these hyperparameters are tuned to preserve fine-grained details, the **resulting images more closely reflect what would have occurred without intervention**. We **show this on the same shoe** image highlighted by the Reviewer, which now retains fine-grained details even more effectively, and can be viewed at [https://anonymous.4open.science/r/figure-800E/b.png](https://anonymous.4open.science/r/figure-800E/b.png).
> > >
> > > In addition, a method’s capabilities are most meaningfully assessed alongside existing approaches targeting similar tasks. Accordingly, we compare our intervention against five alternative methods for color control: prompt-based approaches, best-of-N sampling, ColorPeel [1] (SOTA color control method), ReNO [2] (noise optimization method), and IP-Adapter [3].
> > >
> > > |Color Inj.|ΔE↓|SSIM↑|LPIPS↓|
> > > |-|-|-|-|
> > > |Prompt|31|0.48|0.48|
> > > |BoN=10|24|0.32|0.57|
> > > |BoN=20|23|0.33|0.56|
> > > |BoN=50|21|0.32|0.57|
> > > |ColorPeel|31|0.22|0.66|
> > > |ReNO|27|0.25|0.61|
> > > |IP-Adapter|41|0.29|0.52|
> > > |Ours (local)|**14**|**0.62**|**0.32**|
> > >
> > > We observe that our intervention technique **preserves the image structure more effectively than other color control methods**. Additional qualitative results, available at the provided link, demonstrate that our approach retains fine-grained details significantly better than competing methods.
> > >
> > > **Artifact-free Generation**
> > >
> > > As discussed in the rebuttal, we believe there may have been a misunderstanding regarding Figure 6. For clarity, we provide at the link the image corresponding to our final method in isolation. As shown, there is no blending of the original and target colors along the object boundary: the strawberry is originally red, whereas the final image is entirely blue, with **no residual red**.
> > >
> > > **Significance**
> > >
> > > Considering the significance, it is important to recognize that **color control itself remains critical**—for example, in computational photography, as noted by Reviewer Z14G. Many prior works have attempted to address this challenge [1,2,3] and apply it in high-stakes settings [4,5], where interpretability is essential or even legally required [6,7]. Unlike previous approaches, we propose a **fully interpretable, training-free color control framework that directly meets the demands for trust and reliability.**
> > >
> > > Importantly, by directly intervening in latent space—rather than in the activations of models that operate on it—we introduce a novel approach also well-suited to other evolving architectures. For instance, Representation Autoencoders (RAEs) [8] propose a VAE alternative for flow-matching that leverages CLIP-like vision encoders to construct higher dimensional, semantically rich latent spaces. Similarly, JEPA architectures [9] are gaining traction e.g. for trending world models [10], driven by learning to predict in latent space. Together, these works underscore that **our proposed direct latent interventions have high potential given emerging architectures**.
> > >
> > > Thank you again for your helpful feedback. We hope that, by demonstrating how our method preserves fine-grained details—even on the image you originally highlighted—we have clarified the remaining concerns about its applicability. If so, we would be grateful if you consider increasing the score.
> > >
> > > [1] Butt et al. ColorPeel ECCV’24
> > >
> > > [2] Eyring et al. ReNO NeurIPS’24
> > >
> > > [3] Ye et al. IP-Adapter arXiv’23
> > >
> > > [4] Zhang et al. Pixel super-resolved virtual staining of label-free tissue using diffusion models Nature Com‘25
> > >
> > > [5] Fu et al. Satellite Remote Sensing Grayscale Image Colorization Based on Denoising Generative Adversarial Network. Remote Sens.‘24
> > >
> > > [6] Article 86 of the EU AI Act (2024/1689)
> > >
> > > [7] Artificial Intelligence and Data Act, Canada's Bill C-27 2022
> > >
> > > [8] Zheng et al. Diffusion Transformers with Representation Autoencoders arXiv’25
> > >
> > > [9] Assran et al. Self-Supervised Learning from Images with a Joint-Embedding Predictive Architecture ICCV’23
> > >
> > > [10] Maes et al. LeWorldModel: Stable End-to-End Joint-Embedding Predictive Architecture from Pixels arXiv’26

---

### Official Review · Reviewer_Z14G · 2026-03-12

**Soundness:** 3
**Presentation:** 3
**Significance:** 3
**Originality:** 3
**Overall Recommendation:** 4
**Confidence:** 4

**Summary:**

This paper dives into how FLUX (a cutting-edge text-to-image model that uses flow matching) actually “thinks” about color in its internal VAE latent space. Turns out, even though the latent space is huge and high-dimensional, color isn' t scattered everywhere (it's neatly packed into a tiny 3D corner, which the authors call the Latent Color Subspace). They show this 3D subspace lines up surprisingly well with the familiar HSL color model, and they even trace how colors shift step-by-step along the sampling path.  Using this insight, they design a simple, plug-and-play color editing trick (no extra training, no new parameters, no adapters, no fine-tuning). Just pure math: you can tweak colors globally or locally, right at inference time.

**Compliance With Llm Reviewing Policy:**

Affirmed.

**Key Questions For Authors:**

1. Have you tested the existence of a similar 3D color subspace in other popular T2I models (e.g., Stable Diffusion 3, SDXL, PixArt‑α)? If not, do you have any hypotheses about why FLUX's VAE might exhibit this structure and whether similar insights could be extracted from other architectures?

2. Could you provide a quantitative comparison between your LCS intervention and existing color control methods (e.g., prompt‑based, IP‑Adapter, ControlNet) on a common benchmark? Metrics such as color fidelity (ΔE), structural similarity (SSIM, LPIPS), and inference time would help readers understand the practical advantages and limitations of your approach.

3. Your qualitative examples focus on simple objects and uniform backgrounds. Could you show results on more challenging real‑world photographs (for instance, a portrait with fine hair details, a landscape with vegetation and sky, or a street scene with multiple colorful objects) and discuss any failures or artifacts that arise?

4. how does your color manipulation affect the model's rendering of lighting effects (e.g., shadows, highlights, reflections)? For a photorealistic edit, changing an object's color should also adjust its interaction with light. Do you observe any inconsistencies, and if so, how might they be mitigated?

**Limitations:**

No. Can you quantify the precision of your localized editing? For example, if you provide a mask for an object, what is the IoU of the edited region with the target mask, and how does the color accuracy vary near object boundaries?

**Strengths And Weaknesses:**

Strengths:
1. It's surprising (and useful scientifically) that a modern generative model's VAE organizes colors in such a clean, 3D space that lines up almost perfectly with the HSL color model. For computational photography, this gives us real insight into how these deep models “think” about color and light (opening the door to new editing tools that are easier to understand and more intuitive to control).
2. The method works by tweaking the model's internal latent representations while it's generating, so there's no need to retrain anything or add extra modules. That makes it practical (especially for things like real-time photo editing or running on phones and other devices where speed and simplicity really matter).
3. The paper backs everything up with solid numbers: they track color shifts step-by-step during generation and compare results against professional-grade ground-truth color values using industry-standard color difference metrics (like ΔE). The LCS model nails both prediction  and manipulation (it's accurate).
4. Most of the demos show global color adjustments, but the authors also drop a subtle hint (via spatial masking, see Figure 10) that the same idea could be applied locally. That means you could soon tweak just  one object's color (say, making only the shirt red, not the whole scene), which is exactly what pros do all the time in computational photography.

Weaknesses:
1. The study is conducted exclusively on FLUX.1. Different VAEs (e.g., those in Stable Diffusion, SDXL, or other flow‑based models) may encode color differently. Without cross‑model validation, it is unclear whether the LCS phenomenon is universal or specific to FLUX. This limits the practical impact for the broader computational photography community, which uses a variety of models.

2. The qualitative examples primarily feature isolated objects with uniform colors and simple backgrounds (e.g., a Rubik's cube, a fire truck, plain color patches). Computational photography often deals with complex scenes containing intricate textures, shadows, reflections, and multiple overlapping objects. The method's robustness in such scenarios remains unverified.

3. Color editing in photography is inseparable from illumination. Changing a pixel's hue/saturation/lightness in isolation can produce unrealistic results when lighting is complex (e.g., specular highlights, colored shadows, global illumination). The paper does not explore how the LCS intervention interacts with the model's lighting synthesis (e.g., if we change the color of a shiny metal sphere, do the highlights adjust plausibly)? This is a critical aspect for photorealistic editing.

4. Although local editing is mentioned, the paper lacks a systematic evaluation of how accurately the method can target specific objects without affecting surrounding areas. For photography, this is essential for tasks like selective color grading or object recoloring.

---

> ### Author Rebuttal · Authors · 2026-03-30
>
> We thank the Reviewer for finding our work useful, surprising, efficient, and having solid numerical support. We will add all changes in the final version.
>
> **(W1/Q1) LCS in other models**
>
> We refer the Reviewer to the response to Reviewer aBcj W4. We confirm the existence of a LCS within the VAE spaces of SD3.5, Qwen-Image and FLUX.2 and preliminary intervention capabilities.
>
> **(W3/Q4) Light**
>
> Our intervention performs well at lighting. In Fig. 6 (final method at mid row, t=9), the edited strawberry keeps consistent lighting even after recoloring, including highlights/shadows. Preservation of fine-grained details largely stems from our interpolation strategy. In contrast, the simpler Type I approach degrades details. More examples of complex and realistic lighting are in the Appendix, including Figure 8 row 2 (can has specular highlights) and Figure 9 row 2 (diamond has a color-consistent reflection).
>
> This arises from intervening at the appropriate timesteps to allow the model to make the necessary fine-grained adjustments like shadows/highlights/reflections. Overall, our approach maintains the model’s ability to render these effects.
>
> **(W2/Q3) Complex scenes**
>
> To confirm robustness to prompt complexity, we LLM-rewrite prompts from GenEval’s color task reported in Tab 2 (e.g. “a photo of a blue fire hydrant” → “a photo of a blue fire hydrant on a busy city street with pedestrians, cars, and storefronts in the background”). When applying our local interventions, we achieve 67% accuracy, close to the 70% on the original prompts (3% drop matches the drop observed for the base prompting strategy: 79%→76%).
>
> As discussed in W3/Q4, the base model retains its ability to depict fine-grained textures, shadows, and reflections, and the parrot in Fig 7 shows our method can modify multi-colored objects.
>
> We include qualitative results for longer, more complex prompts with challenging backgrounds inspired by the Reviewer’s Q3 suggestions, along with multi-modification examples, which can be viewed at this anonymous GitHub link [https://anonymous.4open.science/r/figure-800E/f.png](https://anonymous.4open.science/r/figure-800E/f.png). Our approach can handle more intricate prompts and selectively modify target objects without modifying others, even when multiple colorful objects are present (e.g. “A photo of a book laying on bananas, strawberries, and zucchini”). It keeps global coherence by adjusting supporting details, e.g. changing the smoke and reflections in “A photo of a crystal ball on a table in a mysterious library, it releases smoke”. Limitations for these more intricate prompts largely mirror those of their simpler counterparts (see response to Reviewer Mftg Q2).
>
> The examples show we can apply multiple modifications in a single image, even when targets are in close proximity (e.g. dog and cat) or overlapping (e.g. present behind shoe).
>
> **(W4/L1) Local editing**
>
> We split patches into 4 sets based on masks: (1) intervened patches at the boundary, (2) untouched patches adjacent to the intervened, and (3) the remaining intervened and (4) untouched patches. For the Precise (natural) dataset, we compute the average percentage of patches close to the target color (ΔE<25). We rescale based on the percentage of recolored patches under global intervention and before intervention—recognizing that detailing naturally causes deviations and the original image may contain small amounts of the target color—yielding (3) 100%, (1) 73%, (2) 22%, and (4) 5%. This indicates a high recoloring rate in intervention regions (3) and low rates outside (4). As intended, the model smooths transitions and refines shapes at boundaries (2, 3), with comparable deviations from the strict intervention region just inside and outside the boundary.
>
> **(Q2) Quantitative comparisons**
>
> We include Best of N (BoN), which selects the best image from N. We also benchmark IPAdapter; Color Peel [1], the leading color control method with publicly available code; and ReNO [2], a noise-optimization approach. We evaluate a subset of Precise (Natural) limited to 15 colors.
> |Color Inj. |ΔE↓|ΔH↓|ΔS↓|ΔL↓|SSIM↑|LPIPS↓|
> |-|-|-|-|-|-|-|
> |None|47|88|53|19|-|-|
> |Prompt|31|52|31|13|0.48|0.48|
> |BoN=10|24|46|31|10|0.32|0.57|
> |BoN=20|23|48|30|9|0.33|0.56|
> |BoN=50|21|46|30|9|0.32|0.57|
> |ColorPeel|31|68|32|12|0.22|0.66|
> |ReNO|27|34|27|12|0.25|0.61|
> |IPAdapter|41|74|41|18|0.29|0.52|
> |Ours local|**14**|**20**|16|**5**|**0.62**|**0.32**|
> |Ours global|16|34|**15**|8|0.58|0.38|
>
> ColorPeel needs per-color training, and IPAdapters need training on top of the base model. BoN, ReNO need more inference time. Among presented, only None, Prompt and Ours need neither training nor meaningful inference time increase.
>
> Our color interventions are not only more efficient, but also more precise. Notably, our approach is the only one grounded in interpretable principles.
>
> [1] Butt et al. ColorPeel ECCV’24
>
> [2] Eyring et al. ReNO NeurIPS’24

---

> > ### Author Rebuttal · Reviewer_Z14G · 2026-04-05
> >
> > Thanks to the authors for the rebuttal. Some of my issues have been resolved. But I'm still not fully convinced, such as the potential for inconsistencies of lighting. So I'll stick with my original score.

---

> > > ### Author Response · Authors · 2026-04-08
> > >
> > > We are glad we could resolve your other doubts and that your overall impression of our paper remains positive. To address the only remaining uncertainty about the effect of our interventions on lighting, we supplement our earlier provided qualitative examples with new quantitative analysis.
> > >
> > > **Lighting realism confirmed by users**
> > >
> > > To validate lighting realism, we conducted a preliminary user study with non-research-expert participants. We reused 20 pairs of original and color-intervened images containing visible shadows, highlights, and reflections. Ten users were asked to indicate which image (original or intervened) appeared more realistic with respect to: a) shadows, b) highlights, c) reflections, and d) overall lighting.
> > >
> > > ||Original|Intervened|
> > > |-|-|-|
> > > |Shadow|53%|47%|
> > > |Highlight|50%|50%|
> > > |Reflection|48%|52%|
> > > |Overall|55%|45%|
> > >
> > > As shown in the table above, user preferences are nearly evenly split across all four aspects, consistently hovering around 50/50. This suggests that **our interventions have little to no impact on the model’s ability to produce realistic lighting**.
> > >
> > > **Preservation of highlights and shadows**
> > >
> > > We quantitatively assess the preservation of highlights and shadows by comparing their centers in the original and intervened images.
> > >
> > > Shadows and highlights within each object mask are identified using the HSV brightness channel—pixels below the 30th percentile as shadows and above the 95th percentile as highlights. The centers of mass of these regions define vectors from shadow to highlight, and the displacement and angle between these vectors in original and altered images quantify changes in lighting.
> > >
> > > We find that the median of the shadow / highlight center shift is only 3% / 5% of the image width, and that the median angular change is only 9°. This provides **quantitative evidence that the positions and directional relationships of highlights and shadows are largely preserved**.
> > >
> > > **Conclusions**
> > >
> > > Thoroughly quantifying lighting usually requires controlled conditions and accounting for many variables, but our scenes are natural, which makes this challenging. Hence, we compare our images to those before intervention—displaying the lighting quality we aim to preserve—to assess the extent of deviation and determine whether these changes influence user perceptions of realism, finding that the **deviations are minor and do not significantly affect user preferences**.
> > >
> > > Thank you again for your helpful feedback and for finding our work insightful. While our original motivation for the interventions was to confirm the causality of the interpretation model, your suggestions have allowed us to demonstrate our method’s potential as a powerful tool in computational photography. We hope our clarifications address the remaining doubts, and if so, we would be grateful if you might consider updating the score.

---

### Official Review · Reviewer_Mftg · 2026-03-13

**Soundness:** 2
**Presentation:** 2
**Significance:** 2
**Originality:** 2
**Overall Recommendation:** 4
**Confidence:** 3

**Summary:**

This paper explores the internal representation of color in the VAE latent space of the FLUX text-to-image model and proposes the concept of a Latent Color Subspace (LCS). This paper shows that color embeddings form a three-dimensional structure analogous to the Hue–Saturation–Lightness (HSL) color system and derive a mapping between latent coordinates and HSL values using a small set of anchor colors. Also, the authers introduce a method to monitor and manipulate color during generation, demonstrating improved color accuracy and controllability in generated images.

**Compliance With Llm Reviewing Policy:**

Affirmed.

**Key Questions For Authors:**

Generalization across models
Does the proposed Latent Color Subspace structure also appear in other diffusion or flow-matching models (e.g., SDXL, Imagen, or DiT-based architectures)?

Interaction with other semantic attributes
If color is modified in the latent space, does this ever affect other semantic properties such as texture or object identity?

Extension to multi-object scenes
When multiple objects with different colors appear in the same image, how stable is the method when applying localized color interventions?

**Limitations:**

yes

**Strengths And Weaknesses:**

Strengths

1. Interesting mechanistic interpretation of diffusion models
The paper attempts to interpret how semantic attributes (specifically color) are encoded in the latent space of a text-to-image model, which is under-explored in generative models.

2. Simple and training-free method
The proposed method operates directly in the VAE latent space, which makes the method computationally efficient and easy to integrate into existing pipelines.

3. Clear geometric interpretation
The observation that latent color representations form an HSL-like bicone structure provides an intuitive geometric interpretation of color encoding.


Weaknesses

1. Limited analysis
The paper focuses exclusively on color representations in the latent space. While this provides an interesting case study, it is unclear whether the same approach can generalize to other semantic attributes such as shape, texture, or object identity.

2. Empirical rather than theoretical justification
The existence of the Latent Color Subspace is mainly supported by empirical observations. A more rigorous theoretical explanation of why such a structure emerges in the VAE latent space would strengthen the contribution.

3. Evaluation is relatively narrow
The experiments mainly evaluate color accuracy and structural similarity. It would be beneficial to test the method on more diverse datasets or tasks (e.g., real-world editing benchmarks, multi-object scenes).

4. Dependence on anchor color selection
The mapping between the latent space and HSL coordinates relies on a small set of manually selected anchor colors. The robustness of this mapping under different anchor sets or models is not thoroughly investigated.

---

> ### Author Rebuttal · Authors · 2026-03-30
>
> We thank the Reviewer for recognizing the appeal of our mechanistic interpretation in an under-explored area, the simplicity and efficiency of our applicable methodology, and our clear geometric interpretation. We will incorporate all suggested changes in the final version.
>
> **(W2) Theoretical justification**
>
> From a theoretical perspective, VAE latent components are pushed to be orthogonal by the KL term regularizing covariance to stay close to zero and the reconstruction objective [4] (as orthogonality is also necessary for generalization [3]). But VAE space is low-dimensional (e.g. 16 dimensions), which creates a low budget for representing patches. Together, this creates conditions where information is likely to be encoded with frequently used, very generalizable features, which typically end up being patch-level structural details like color, edges, and basic patterns.
>
> We see these factors in the LCS: color is largely separated from texture, lightness is assigned its own dimension, the smooth transitions of the circular encoding of hue promote generalizability, and saturation is condensed into the same dimensions as hue by assigning it to the previously unused radial component.
>
> **(W1) Extension to non-color subspaces**
>
> As discussed in W2, if other interpretable subspaces exist, they likely capture low-level features (e.g. edges) rather than high-level semantic information like object identity, and operate on individual patches too small to encode larger structures like shapes. In this sense, similar to the LCS, they would correspond to a single patch's structure.
>
> We preliminarily explored these directions by modifying individual latent channels of encoded images and visualizing the decoded results, observing changes such as structural modifications and color gradients consistent across patches that support this theory. We find color to be a sufficiently coherent attribute to study on its own, as it is intuitive—closely mirroring the well-studied HSL color model—and offers immediate practical applications. This intuition does not clearly extend to structural features that contribute to texture, which require different considerations for effective use. We focus on color comprehensively, with the hope of inspiring future work to investigate other subspaces with the same level of depth.
>
> **(W3 / Q3) Real-world applications of the LCS and multi-object scenes**
>
> We refer the Reviewer to our response to Reviewer Z14G W2 / Q3. We observe that our intervention successfully extends to prompts with complex backgrounds, multiple objects, and interventions, and that edits to complex prompts remain similarly stable to simpler ones.
>
> **(W4) Anchor selection**
>
> We investigate the method’s sensitivity to anchor selection by validating under two settings: random re-assignment of the 6 hue anchors (averaged over three random hue-selection seeds), and extension to 24 hue anchors (as denser coverage may improve accuracy). Below, we report observation via CIEDE2000 (ΔE) at timestep 50 in the Objects setting (as in Table 1), computed both per pixel and for the average pixel.
>
> |# anchors|ΔE (per pix)|ΔE (avg)|
> |-|-|-|
> |6 (orig)|14|10|
> |6 (rand)|14|10|
> |24|13|9|
>
> We observe that resampling the anchors has negligible impact on the accuracy of the LCS and increasing the number of anchors fourfold yields a reduction in error. Nevertheless, the reduction is small, suggesting that six anchors are enough to achieve highly accurate predictions.
>
> **(Q1) LCS in other models**
>
> We refer the Reviewer to the response to Reviewer aBcj W4. We confirm the existence of a LCS within the VAE spaces of SD3.5, Qwen-Image, and FLUX.2.
>
> **(Q2) Affect on other semantics**
>
> We find that the impact on other semantics is minimal, as evidenced both qualitatively (Figure 7) and quantitatively (Table 3). In some cases (e.g., the shoe in Figure 7), small semantic changes may appear. However, these are not caused by the intervention itself; rather, they arise from modifying the image mid-generation. This process allows the model to make fine-grained adjustments that can improve realism like lighting, blending, and overall coherence. While it may introduce minor shifts driven by model biases (e.g., red shoes being rendered as more sporty than dressy), these may also be desirable: e.g., turning a tree stump green can give the impression that it is covered in moss, and shifting a steak’s color toward red can make it appear more raw (images can be viewed via this anonymous GitHub link [https://anonymous.4open.science/r/figure-800E/f.png](https://anonymous.4open.science/r/figure-800E/f.png)). As shown in Figure 6 at timestep 50 when the model can no longer edit the image, LCS manipulation alone produces minimal semantic changes.
>
> [3] Uselis et al. Compositional Generalization Requires Linear, Orthogonal Representations in Vision Embedding Models arXiv’26
>
> [4] Bhowal et al. Why do Variational Autoencoders Really Promote Disentanglement? PMLR’24

---

> > ### Author Rebuttal · Reviewer_Mftg · 2026-04-04
> >
> > Thanks to the authors for the rebuttal. Although many of my issues have been resolved, the potential for generalizing color disentanglement and its overall impact remain questionable. I will keep my current rating.

---

> > > ### Author Response · Authors · 2026-04-08
> > >
> > > We are glad to have resolved almost all of your doubts and to learn that your overall assessment is positive, given your current score. We are happy to elaborate on the last point: *the potential for generalizing color disentanglement and its overall impact.*
> > >
> > > While generalizing LCS beyond color is undoubtedly interesting, it is important to recognize that **color control itself remains critical**—for example, in computational photography, as noted by Reviewer Z14G. Many prior works have attempted to address this challenge [1,2,3] and apply it in high-stakes settings [4,5], where interpretability is essential or even legally required [6,7]. Unlike previous approaches, we propose a **fully interpretable, training-free color control framework that directly meets the demands for trust and reliability.**
> > >
> > > Importantly, by directly intervening in latent space—rather than in the activations of models that operate on it—we introduce a novel approach also well-suited to other evolving architectures. For instance, Representation Autoencoders (RAEs) [8] propose a VAE alternative for flow-matching that leverages CLIP-like vision encoders to construct higher dimensional, semantically rich latent spaces. Similarly, JEPA architectures [9] are gaining traction e.g. for trending world models [10], driven by learning to predict in latent space. Together, these works underscore that **our proposed direct latent interventions have high potential given emerging architectures**.
> > >
> > > Thank you again for your helpful feedback and for finding our work interesting. We hope our clarifications help highlight the value of our paper, and if so, we would be grateful if you consider increasing the score.
> > >
> > > [1] Butt et al. ColorPeel ECCV’24
> > >
> > > [2] Eyring et al. ReNO NeurIPS’24
> > >
> > > [3] Ye et al. IP-Adapter arXiv’23
> > >
> > > [4] Zhang et al. Pixel super-resolved virtual staining of label-free tissue using diffusion models Nature Com‘25
> > >
> > > [5] Fu et al. Satellite Remote Sensing Grayscale Image Colorization Based on Denoising Generative Adversarial Network. Remote Sens.‘24
> > >
> > > [6] Article 86 of the EU AI Act (2024/1689)
> > >
> > > [7] Artificial Intelligence and Data Act, Canada's Bill C-27 2022
> > >
> > > [8] Zheng et al. Diffusion Transformers with Representation Autoencoders arXiv’25
> > >
> > > [9] Assran et al. Self-Supervised Learning from Images with a Joint-Embedding Predictive Architecture ICCV’23
> > >
> > > [10] Maes et al. LeWorldModel: Stable End-to-End Joint-Embedding Predictive Architecture from Pixels arXiv’26

---

### Official Review · Reviewer_uXQQ · 2026-03-18

**Soundness:** 3
**Presentation:** 3
**Significance:** 2
**Originality:** 3
**Overall Recommendation:** 4
**Confidence:** 4

**Summary:**

This paper identifies a color-related subspace in the VAE latent space and analyzes it. In particular, it proposes a method for manipulating color in latent space using an HSL-based formulation. The paper introduces a methodology for analyzing the Latent Color Subspace (LCS), decoding and encoding it through HSL, and using this representation to infer color during generation and intervene on it in the middle of the process. In other words, it proposes a method that applies shifts in LCS to induce corresponding shifts in HSL space.

**Compliance With Llm Reviewing Policy:**

Affirmed.

**Final Justification:**

The authors’ rebuttal addressed most of my main concerns clearly and satisfactorily. I believe this paper has a merit to be accepted.

The work remains strong in terms of technical and overall contribution. At the same time, when I consider the broader impact of the paper, as well as the extent to which it opens up additional future research directions, I do not think my assessment enough to warrant a score increase. For that reason, I am maintaining my original score.

**Key Questions For Authors:**

Is it possible to change the color of only a specific region rather than the entire image? If so, how well does the method work in that setting?

**Limitations:**

yes

**Strengths And Weaknesses:**

The paper is very intuitive and well written. First, the overall flow is easy to follow and the ideas are presented clearly. Although the method itself is relatively simple, I think the work is meaningful because it is rare to see papers that explicitly analyze a color-related subspace in latent space. The evaluation metrics used to support the claims also make sense and are well chosen.

Weaknesses:
One limitation is that it would be helpful to see experiments beyond HSL, such as with other color systems including HSV, HSB, Lab, or HCL, to better understand how general the proposed idea is across different color representations. In addition, it would be interesting to evaluate whether the method is compatible with additional techniques designed to preserve shape or structure, such as attention-based insertion mechanisms or related interventions.

---

> ### Author Rebuttal · Authors · 2026-03-30
>
> We thank the Reviewer for recognizing the meaningfulness and uniqueness of our work, our well-chosen evaluation metrics, and the clarity of the manuscript. We will incorporate all suggested changes in the final version.
>
>
>
> **(W1) Other Color Representations**
>
> The LCS, which is the VAE’s internal representation of color, emerges naturally during training and remains fixed thereafter. We find that its structure closely resembles the HSL color space, which we therefore use as an approximation for both observation and intervention. Because analytical conversions exist between all considered color spaces, we can map colors from any of these representations into HSL to interface with the LCS. This ensures that our approach is not constrained by a particular choice of color space, as the representations are equivalent. To further assess the precision of interventions across different representations, we also report the resulting changes in components from alternative color spaces (La\*b\* and HCL), including ΔH, ΔC, and ΔL from HCL, as well as Δa* and Δb* from La\*b\*.
>
> |     |ΔH(0-100)|ΔC(0-180)|ΔL(0-100)|Δa*(0-255)|Δb*(0-180)|
> |---------|---------|---------|---------|----------|----------|
> |No color |83       |47       |27       |44        |46        |
> |Prompt   |32       |35       |17       |31        |28        |
> |Local    |**16**       |30       |**11**       |23        |22        |
> |Global   |17       |**28**       |15       |**22**        |**20**        |
>
>
>
> These metrics provide a more holistic view of color accuracy, reinforcing the original findings that our method achieves the target color more closely than prompting alone.
>
>
>
> **(W2, Q) Compatibility with other techniques and regional changes**
>
> One key advantage of the LCS is that it arises naturally within the representation space. Although other adaptation techniques, such as shape and structure preservation, did not explicitly recognize the existence of the LCS, it already formed a subspace within the VAE’s representation space. As long as these methods did not significantly alter that space—which is rarely the case—their effectiveness depended on implicit meaningful use of the LCS.
>
>
>
> Moreover, our proposed approach enables a single, direct intervention in the latent space to produce an equivalent representation with modified color. Importantly, it requires no changes to the architecture or to the semantics of the representations. As a result, the intervention remains compatible with other methods, whether they operate via attention mechanisms or introduce additional modifications to the latent space.
>
>
>
> Indeed, we already validate compatibility with an attention-based mechanism in the context of local edits. This is also where we demonstrate that it is possible to change the color of a specific region rather than the entire image. Specifically, we leverage semantic segmentation derived from attention maps to generate masks for targeted editing. These masks integrate with our intervention approach, as shown in Table 2 (Ours_local) and Figure 7, enabling precise and accurate color modifications of specific objects.

---

> > ### Author Rebuttal · Reviewer_uXQQ · 2026-04-03
> >
> > Thank you to the authors for their thoughtful rebuttal. My concerns have been addressed.

---

> > > ### Author Response · Authors · 2026-04-08
> > >
> > > We sincerely thank you for your thoughtful feedback. We are pleased that our rebuttal has fully resolved your concerns. As indicated in the selected option, we hope that you may increase your score to reflect the satisfaction with our response.

---

### Decision · Program_Chairs · 2026-04-30

**Decision:**

Accept (regular)

**Comment:**

This paper identifies a structured latent colour subspace in the VAE latent space of FLUX, aligned with HSL color model and propose training free closed form colour control method. The paper received three weak accepts and one weak reject, with the rebuttal comprehensively addressing major concerns across all reviews. The three accepting reviewers maintained their scores despite acknowledging resolution of their concerns. The other review did not give enough justification for strong rejection. Given the Novelty of the mechanistic finding, strength of the practical contribution, quality of the rebuttal, I recommend acceptance with the expectation that the authors incorporate the promised presentation improvements, extended comparisons and cross model results into their final version.